# POSITIONAL ENCODING FIELD

**Yunpeng Bai**[*]
University of Texas at Austin
byp215@utexas.edu

**Haoxiang Li**
Pixocial Technology
haoxiang.li@pixocial.com

**Qixing Huang**
University of Texas at Austin
huangqx@cs.utexas.edu

## ABSTRACT

Diffusion Transformers (DiTs) have emerged as the dominant architecture for visual generation, powering state-of-the-art image and video models. By representing images as patch tokens with positional encodings (PEs), DiTs combine Transformer scalability with spatial and temporal inductive biases. In this work, we revisit how DiTs organize visual content and discover that patch tokens exhibit a surprising degree of independence: even when PEs are perturbed, DiTs still produce globally coherent outputs, indicating that spatial coherence is primarily governed by PEs. Motivated by this finding, we introduce the Positional Encoding Field (PE-Field), which extends positional encodings from the 2D plane to a structured 3D field. PE-Field incorporates depth-aware encodings for volumetric reasoning and hierarchical encodings for fine-grained sub-patch control, enabling DiTs to model geometry directly in 3D space. Our PE-Field–augmented DiT achieves state-of-the-art performance on single-image novel view synthesis and generalizes to controllable spatial image editing.

## 1 INTRODUCTION

Diffusion Transformers (DiTs) (Peebles & Xie, 2023) have rapidly emerged as the dominant architecture in visual generation, forming the backbone of recent state-of-the-art image and video models such as Flux.1 Kontext (Labs et al., 2025), Qwen-Image (Wu et al., 2025a), CogVideo (Yang et al., 2024), and Wan (Wan et al., 2025). By encoding images into sequences of patch tokens and applying 2D positional encodings (PEs) (Vaswani et al., 2017), DiTs leverage the scalability of Transformers while preserving the spatial inductive biases necessary for visual synthesis. This design has enabled remarkable progress, supporting high-fidelity image generation and temporally coherent video synthesis (where additional temporal PEs are employed).

Despite their empirical success, the internal mechanisms by which DiTs organize and compose visual content remain relatively underexplored. In this work, we begin with a simple yet striking observation: patch tokens in DiTs exhibit a surprising degree of independence. When positional encodings are reassigned, the model still produces globally coherent output, though with patches reorganized according to the altered PEs. This suggests that spatial coherence in DiTs is primarily enforced by positional encodings rather than by explicit token-to-token dependencies and that manipulating PEs alone can induce structured reconfiguration of spatial content. This property offers a new avenue for spatially controllable generation, where images can be reorganized according to PEs transformation without modifying the token content itself.

Building on this insight, we focus on single-image novel view synthesis (NVS) and extend the positional encodings of DiTs beyond the 2D image plane into a structured 3D field, which we term the Positional Encoding Field (PE-Field). The PE-Field introduces two key innovations: First, we extend standard 2D RoPE (Su et al., 2024) to a 3D depth-aware encoding, embedding tokens in a volumetric field that supports reasoning across viewpoints. Second, we design a hierarchical scheme that subdivides tokens into finer sub-patch levels, allowing different sub-vectors to capture

---

[*]This work was conducted during an internship at Pixocial Technology.

spatial information at varying granularities. Together, these designs transform DiTs into a geometry-aware generative framework that reasons directly in a 3D positional encoding field. As a result, our approach achieves state-of-the-art results in novel view synthesis (NVS) from a single image, and naturally generalizes to spatial editing tasks, where manipulating the PE-Field enables structured control of image content at both global and local levels.

Our contributions are as follows: **1)** We show that DiTs can reorganize image content purely through positional encodings, revealing a previously underexplored property that enables structured spatial editing. **2)** We introduce a depth-augmented positional encoding field that embeds tokens into a 3D space, enabling volumetric reasoning and geometric consistency. **3)** We extend DiTs with multi-level positional encodings, allowing fine-grained spatial control at sub-patch granularity. **4)** Our PE-Field–augmented DiT achieves state-of-the-art results on novel view synthesis (NVS) from a single image, and further generalizes to spatial image editing tasks.

## 2 RELATED WORKS

### 2.1 NOVEL VIEW SYNTHESIS

Novel view synthesis (NVS) is a widely studied and discussed problem which can be broadly divided into two categories: methods based on multiple input images and those based on a single input image. In this work, we focus on the latter. The simplest approach is to directly use a feed-forward model (Hong et al., 2024; Jin et al., 2025) to generate novel views from an input image. Such methods typically rely on learning intermediate, general 3D representations from data. For example, early works adopt multi-plane representations (Zhou et al., 2018; Han et al., 2022; Tucker & Snavely, 2020), PixelNeRF (Yu et al., 2021) employs NeRF (Mildenhall et al., 2020) as the 3D representation, LRM (Hong et al., 2024) uses tri-plane representations, and 3D-GS (Kerbl et al., 2023) has also been adopted by methods such as PixelSplat (Charatan et al., 2024). Other methods (Wiles et al., 2020; Rombach et al., 2021; Rockwell et al., 2021; Park et al., 2024) incorporate additional results from monocular reconstruction to provide an explicit geometric structure, where warping into the target view is used which is then followed by inpainting to synthesize novel views.

Recently, with the breakthrough of diffusion-based generative models, an increasing number of works have investigated the use of diffusion models for NVS, including GeNVS (Chan et al., 2023), Zero-1-to-3 (Liu et al., 2023), ZeroNVS (Sargent et al., 2024), and CAT3D (Gao et al., 2024; Wu et al., 2025b). However, directly encoding camera pose conditions as text embeddings makes it difficult to precisely control viewpoint changes. Reconfusion (Wu et al., 2024) uses PixelNeRF (Yu et al., 2021) features as diffusion conditions, but consistency across views cannot be guaranteed. The paradigm of monocular reconstruction followed by warping and inpainting has also been adopted in diffusion-based methods (Zhang et al., 2024; Chung et al., 2023; Shriram et al., 2024; Yu et al., 2024; Cao et al., 2025), where diffusion is used for the inpainting stage. However, reprojection errors in the warped image may disrupt the semantics of the source image and are difficult to correct during inpainting. To address this issue, GenWarp (Seo et al., 2024) proposes to use warped 2D coordinates as input instead of directly warping the image, and this idea has been extended to videos in later work (Seo et al., 2025). However, since view transformation inherently occurs in 3D space, relying solely on 2D coordinates remains ambiguous, and these methods require training additional branches to handle coordinate input. Many video-based models (Sun et al., 2024; Huang et al., 2025; Chen et al., 2025; Ren et al., 2025; Zhang et al., 2025; Song et al., 2025; Liang et al., 2025) incorporate camera control to achieve NVS, but when only the target view is required, generating intermediate frames is unnecessary. CausNVS (Kong et al., 2025) also explores an autoregressive approach for novel view synthesis.

### 2.2 DITS FOR IMAGE GENERATION AND EDITING

Diffusion Transformers (DiTs) were first introduced by (Peebles & Xie, 2023), who replaced the commonly used U-Net backbone in diffusion models (Rombach et al., 2022) with a pure Transformer architecture. This design leveraged the scalability and flexibility of Transformers while retaining the generative power of diffusion, and has since become the foundation of many state-of-the-art image and video generation models. Building on DiT, subsequent works such as Stable Diffusion 3 (SD3) (Esser et al., 2024), Flux.1 Kontext (Labs et al., 2025), Qwen-Image (Wu et al.,

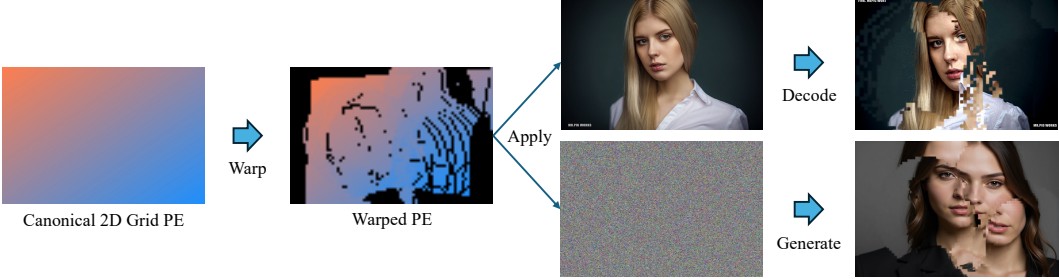

Figure 1: Illustration of DiT patch-level independence. When positional encodings (PEs) of image tokens or noise tokens are reassigned, the decoded or generated outputs still produce semantically meaningful images. The resulting structures follow the positional encoding reassignment, while boundaries between patches remain visually distinct.

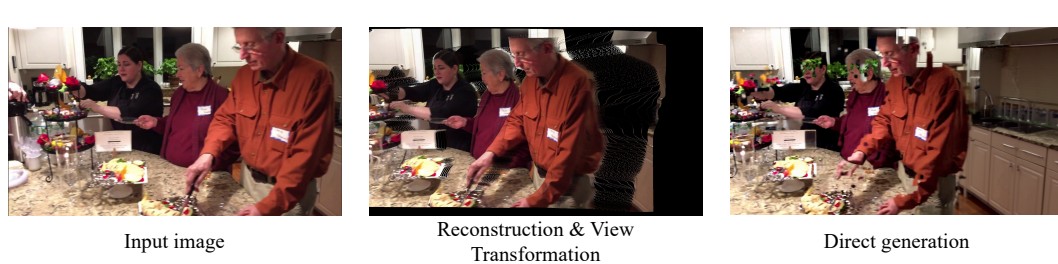

| Input image | Reconstruction & View Transformation | Direct generation |

Figure 2: Illustration of our direct novel view synthesis (NVS) Results. We apply 2D positional encodings (PEs) derived from 3D reconstruction and view transformation directly to the source-view image tokens. Using these modified tokens as image conditions in DiT enables direct generation of a relatively accurate novel-view image.

2025a), CogVideo (Yang et al., 2024), and Wan (Wan et al., 2025) have established DiT as the main backbone for large-scale generative modeling. Owing to its flexible architecture, DiT can be naturally extended by incorporating the tokens of a context image directly into the input sequence, enabling end-to-end image editing within the same generative framework. This simple yet effective strategy has been widely adopted in current mainstream editing models (Labs et al., 2025; Wu et al., 2025a), demonstrating the versatility of DiTs for controllable generation tasks. In contrast, we propose equipping DiTs with a 3D-aware hierarchical positional encoding field, enabling controllable and geometry-aware generation and editing solely through transformations on positional encodings.

## 3 METHOD

### 3.1 TOKEN MANIPULATION FOR VIEW SYNTHESIS

**Patch-level independence in DiT-based generative models.** DiT-based architectures model image generation by patchifying the input and representing each patch as a token with a 2D positional encoding (PE). While tokens collectively reconstruct the image, we find that each token mainly encodes its local patch and retains a degree of independence. As shown in Figure 1 (Top), reassigning tokens' PEs leads to images reorganized according to the new layout, with clear patch boundaries indicating independent decoding. This independence also appears during denoising: as shown in Figure 1 (Bottom), reassigning PEs of noise tokens still yields globally coherent results (e.g., a face) but with block-wise discontinuities aligned with the modified positions. These findings suggest that global coherence is largely enforced by PEs, enabling the possibility of spatial editing by manipulating token positions through their PEs without altering token content.

**Towards novel view synthesis via token manipulation.** In this work, we mainly want to leverage these findings to address novel view synthesis (NVS) problem from a single image. A straightforward solution is to perform single-view 3D reconstruction followed by view transformation and

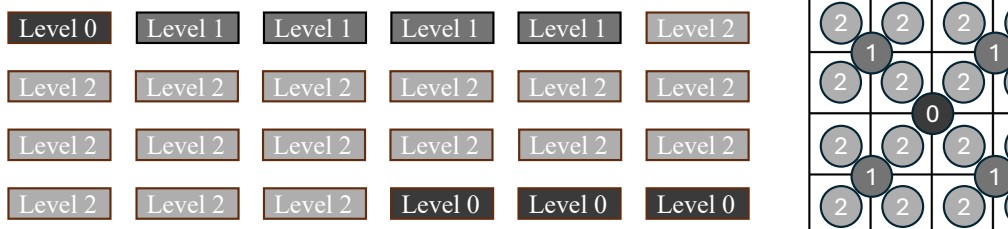

Figure 3: Illustration of hierarchical RoPE allocation in Flux (24 heads). Each rectangle on the left represents the subvector computed by one head, with colors indicating the RoPE level. Black denotes the original patch-level RoPE ($l = 0$), covers a 256 pixels patch. Level $l = 1$ corresponds to 64 pixels, and level $l = 2$ to 16 pixels. The square on the right represents a patch corresponding to one token, illustrating how different levels of positional encodings map to their respective 2D spatial locations, where $l = 2$ corresponds to a $1/16$-sized patch.

inpainting, but this pipeline is often prone to errors (Seo et al., 2024). Instead, we directly manipulate DiT's image token positions: conditioned on the source reconstruction and target camera pose, we reassign positional encodings so that tokens migrate to their new projected locations. This allows recomposing image content under novel viewpoints within the DiT generative process, avoiding errors from direct image-space warping. As shown in Figure 2, this approach demonstrates a partial but effective ability to perform NVS, but artifacts remain due to: (1) resolution mismatch—positional grids from patch tokens (e.g., $16 \times 16$ pixels) are coarser than dense 3D reconstructions, limiting alignment precision. The manipulation can only rearrange image content at the patch level, but it cannot alter the content within each patch. and (2) depth ambiguity—multiple 3D points may project to the same token location. Without explicit mechanisms to disambiguate depth, generated tokens can collapse into inconsistent local structures. To adapt DiTs for NVS through positional encoding transformations, we introduce two key modifications to the existing PE design, extending it into a structured 3D field representation.

### 3.2 MULTI-LEVEL POSITIONAL ENCODINGS FOR SUB-PATCH DETAIL MODELING

In the current DiT architecture, each image patch is represented as a single token, i.e., a one-dimensional vector $\mathbf{x}_i \in \mathbb{R}^d$, which is fed into the transformer for computation. Within the transformer, multi-head self-attention (MHA) is applied by projecting $\mathbf{x}_i$ into multiple subspaces (heads), $h \in \{1, \ldots, H\}$ with per-head dimension $d_h$ (typically $d_h = d/H$) enabling the model to capture diverse relationships across tokens. Current mainstream DiT models, such as Flux and SD3, first obtain queries, keys, and values by linear projections of the hidden states: $Q = XW_Q, K = XW_K, V = XW_V, X \in \mathbb{R}^{B \times T \times d}$. The results are then reshaped into $H$ heads with per-head dimension $d_h = d/H$: $Q, K, V \in \mathbb{R}^{B \times T \times d} \to \mathbb{R}^{B \times H \times T \times d_h}$. For each head, attention is computed as $\text{head}^{(h)} = \text{softmax}\left(\frac{Q^{(h)} K^{(h)\top}}{\sqrt{d_h}}\right) V^{(h)}$. Finally, the outputs of all heads are concatenated and projected back to dimension $d$. However, all heads share the same positional encodings (specifically RoPEs (Su et al., 2024)), which are tied to patch-level locations. Thus, although each token is divided across multiple heads for modeling, it still encodes the holistic content of an entire patch, without explicitly capturing finer-grained details within the patch.

We argue that this design limits the transformer's ability to capture sub-patch structures that are crucial for tasks involving fine spatial transformations, such as novel view synthesis. Our goal is not to discard the different correspondences already learned by different heads at the patch level, but rather to enrich them with intra-patch detail modeling. To this end, we build directly on the head-splitting structure of MHA, augmenting it with multi-level hierarchical positional encodings so that each head's subspace captures not only patch-level information but also finer-grained details, while remaining highly compatible with the original architecture since the finer-level PEs differ little from the original ones.

Concretely, we retain a subset of heads that use the original patch-level RoPE ($l_h = 0$) to preserve the pretrained global structure, while other heads adopt finer-grained RoPEs derived from higher

resolution grids (see Figure 3). At level $l_h = 0$, each positional encoding corresponds to the original patch-level RoPE (e.g., one token covers $16 \times 16$ pixels). When moving to higher levels, the positional grid resolution is increased: each step doubles the resolution along both axes, so the effective cell size shrinks by a factor of 2 per axis (i.e., by 4 in area). Let $\{\text{RoPE}^{(l_h)}\}_{l_h=0}^{M-1}$ denote the hierarchy of positional encodings, where larger $l_h$ corresponds to higher spatial resolution (doubling per axis per level). Queries and keys in head $h$ are rotated by the level-specific RoPE: $\mathbf{Q}_h = \text{RoPE}^{(l_h)}(Q^{(h)}), \mathbf{K}_h = \text{RoPE}^{(l_h)}(K^{(h)})$. We automatically choose the number of levels $M$ from the total number of heads $H$ in the pretrained architecture:

$$M = \left\lfloor \log_4(3H + 1) \right\rfloor, \qquad W = \frac{4^M - 1}{3},$$

where $W$ is the cumulative geometric series $1 + 4 + \cdots + 4^{M-1}$, which represents the total number of hierarchical heads that can be accommodated under the current architecture. Each head index $h \in \{1, \ldots, H\}$ maps directly to a level via the rule that exactly matches the geometric quotas $1\!:\!4\!:\!16\!:\cdots$ whose total sums to $W$, and falls back to the original RoPE ($l = 0$) for surplus heads:

$$l_h = \begin{cases} \left\lceil \log_4\big(3h + 1\big) \right\rceil - 1, & h \leq W, \\ 0, & h > W, \end{cases} \quad \text{clipped to } [0, M-1].$$

Any heads beyond the geometric budget $W$ default to $l = 0$ to minimize disruption of pretrained patch-level priors. Taking Flux as an example, we divide each sub-vector into three levels: In Flux, there are 24 heads in total. The first head corresponds to $l = 0$, i.e., the original patch-level RoPE. Heads 2–5 are assigned to $l = 1$, and heads 6–21 to $l = 2$. The remaining heads 22–24 cannot be allocated under this scheme and are therefore reassigned back to $l = 0$. As illustrated in Figure 3, different colors indicate different PE levels. The coarsest level corresponds to a $16 \times 16$-pixel patch, while the finest level corresponds to a $4 \times 4$-pixel patch. This hierarchical design enables flexible spatial transformations: direct manipulations of sub-patch RoPE yield local geometric adjustments in the reconstruction while preserving pretrained patch-level correspondences.

## 3.3 Depth-aware rotary positional encoding

In standard 2D RoPE, the horizontal ($x$) and vertical ($y$) coordinates are encoded independently. Each axis is assigned a dedicated subspace of the embedding vector, within which a 1D RoPE is applied. Concretely, the token vector is partitioned into two segments, one modulated by the RoPE corresponding to the horizontal coordinate $x$ and the other by the RoPE for the vertical coordinate $y$. This factorized scheme ensures that the dot product of two rotated queries and keys encodes relative displacements along both axes, while keeping the rotations invertible and dimensionally consistent.

To allow DiT to leverage positional encodings for reasoning about depth relationships between tokens that overlap in the 2D projection, following the above principle, we extend RoPE to include a third spatial axis for depth, which refers to the distance of each pixel's corresponding 3D point from the camera along the optical axis (that is, its z coordinate in the camera coordinate system). In addition to the subspaces for $(x, y)$, we introduce another subspace for the depth $z$. Each coordinate $(x, y, z)$ thus has its own 1D RoPE encoding, applied to a disjoint part of the embedding vector:

$$\mathbf{Q}^{(h)} = \big[\, \text{RoPE}_x^{(l_h)}(\mathbf{Q}_x^{(h)}),\ \text{RoPE}_y^{(l_h)}(\mathbf{Q}_y^{(h)}),\ \text{RoPE}_z^{(l_h)}(\mathbf{Q}_z^{(h)}) \,\big],$$

$$\mathbf{K}^{(h)} = \big[\, \text{RoPE}_x^{(l_h)}(\mathbf{K}_x^{(h)}),\ \text{RoPE}_y^{(l_h)}(\mathbf{K}_y^{(h)}),\ \text{RoPE}_z^{(l_h)}(\mathbf{K}_z^{(h)}) \,\big],$$

where $\mathbf{Q}_x^{(h)}, \mathbf{Q}_y^{(h)}, \mathbf{Q}_z^{(h)}$ (and $\mathbf{K}_x^{(h)}, \mathbf{K}_y^{(h)}, \mathbf{K}_z^{(h)}$) denote the corresponding vector segments allocated to each axis. This extension yields a 3D spatial RoPE that encodes relative offsets not only in the image plane but also along the depth axis, enabling the Transformer to model volumetric correspondences and maintain geometric consistency across viewpoints.

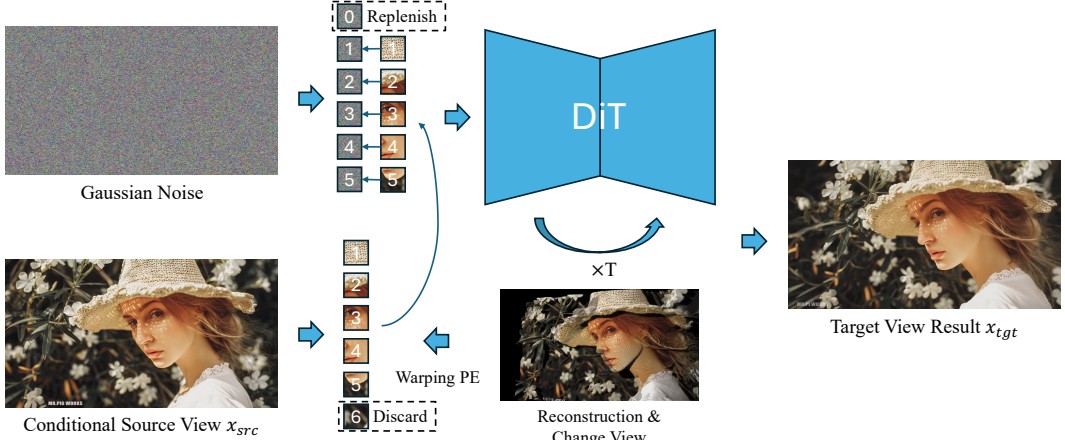

Figure 4: The transformer takes both noise tokens and source-view image tokens. Noise tokens are placed on a 2D grid with depth set to zero, while image tokens are assigned hierarchical PEs according to their projected positions from monocular reconstruction and view transformation, with depth values taken from the reconstruction. Tokens projected outside the grid (e.g., index 6) are discarded, and empty grid locations without image tokens (e.g., index 0) are filled by noise, which is refined to generate plausible content.

## 3.4 OVERALL ARCHITECTURE AND TRAINING OBJECTIVE

These two components together form a new 3D field–based positional encoding, which we apply to the DiT architecture to jointly process noise tokens and source-view image tokens, resulting in our NVS-DiT model. As illustrated in Figure 4, noise tokens are placed on a regular 2D grid with depth initialized to zero, while source-view image tokens are projected into the target camera view via monocular reconstruction and view transformation. Each image token is assigned a hierarchical 3D positional encoding $(x, y, z)$ that captures its detailed target spatial location and depth. Tokens projected outside the valid grid are discarded, and empty positions are filled with noise tokens, which are progressively refined by the transformer to generate geometrically consistent content. This design enables the model to integrate observed image evidence with generative completion, achieving novel view synthesis within the DiT framework.

To train the model, we leverage multi-view supervision under a rectified-flow (Liu et al., 2022) objective. Specifically, we adopt the rectified flow–matching loss:

$$\mathcal{L}_\theta = \mathbb{E}_{t \sim p(t),\, x_{tgt},\, x_{src}^{trans-PE}} \left[ \left\| v_\theta(z_t, t, x_{src}^{trans-PE}) - (\varepsilon - x_{tgt}) \right\|_2^2 \right],$$

where $x_{src}^{trans-PE}$ and $x_{tgt}$ denote the image tokens of the source view with transformed PEs and the target view, respectively, obtained by the corresponding DiT's VAE encoder. $z_t$ is the linearly interpolated latent between clean latent $x_{tgt}$ and Gaussian noise $\varepsilon \sim \mathcal{N}(0, 1)$, defined as $z_t = (1 - t)x_{tgt} + t\varepsilon$.

## 4 EXPERIMENTS

### 4.1 IMPLEMENTATION DETAILS

Our model is built on Flux.1 Kontext (Labs et al., 2025), which generates images conditioned jointly on a text prompt and a reference image. This architecture naturally aligns with our design, as it already integrates reference-image tokens, providing a seamless foundation for incorporating our PE-Field framework. We remove its text input and condition solely on the reference image. To train our NVS model, we use two multi-view datasets, DL3DV (Ling et al., 2024) and MannequinChallenge (Li et al., 2020), both processed with VGGT (Wang et al., 2025) to obtain per-image depth maps and corresponding camera poses. Additional details are provided in the **Appendix**.

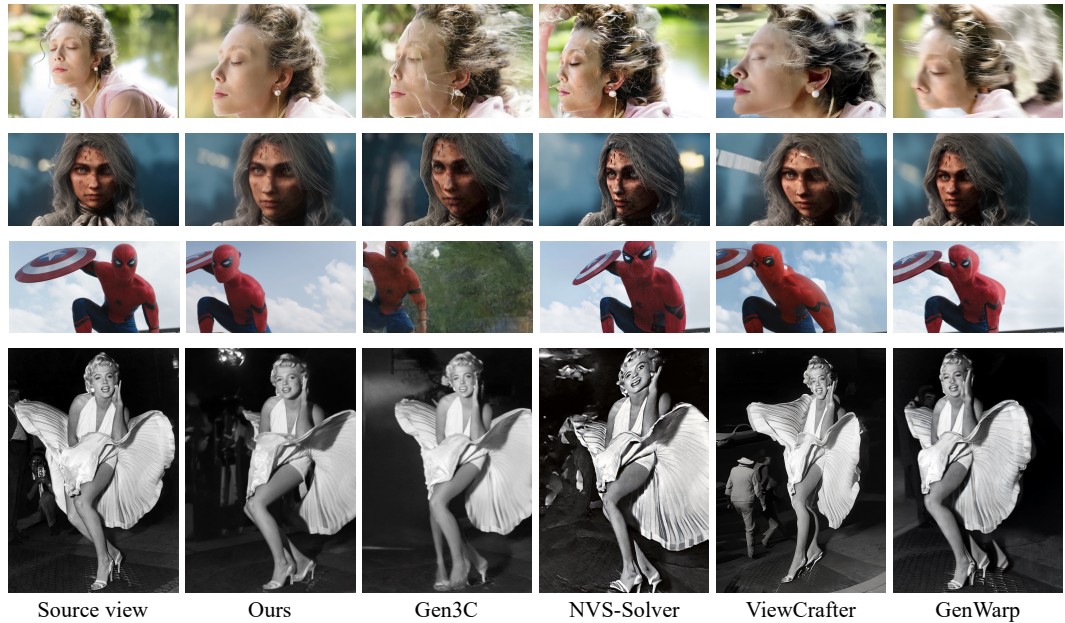

| Source view | Ours | Gen3C | NVS-Solver | ViewCrafter | GenWarp |

Figure 5: Visualization of novel view synthesis results where the source image (left) is rotated 30° to the right. Compared with other methods, our approach achieves accurate viewpoint transformation while preserving consistency with the source image and avoiding noticeable artifacts.

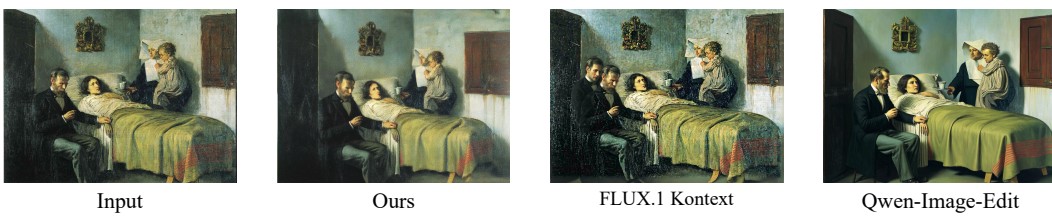

| Input | Ours | FLUX.1 Kontext | Qwen-Image-Edit |

Figure 6: Comparison with prompt-based image editing methods. Our approach enables accurate control of rotation angles while maintaining consistency with the input image.

## 4.2 COMPARISONS WITH RELEVANT METHODS

We mainly compare our approach with several baseline methods (listed in Table 1) in the single-image novel view synthesis setting. Experiments are conducted on three datasets, Tanks-and-Temples (Knapitsch et al., 2017), RE10K (Zhou et al., 2018), and DL3DV (Ling et al., 2024). In each case, a single input image is provided, and subsequent frames are generated under different target viewpoints. For methods that require depth or point cloud as conditional input, we uniformly use the predictions obtained from VGGT as input. We then calculated three metrics, PSNR, SSIM (Wang et al., 2004), and LPIPS (Zhang et al., 2018), and reported the average scores for all test samples in Table 1. Our method outperforms existing approaches across all metrics on all three datasets. Qualitative comparison with a subset of representative methods is presented in Figure 5. We observe that GEN3C often propagates reconstruction artifacts into the final results, leading to noticeable white streaks and irregular boundaries. NVS-Solver and ViewCrafter tend to introduce depth-warping errors, which negatively affect the geometric accuracy of the synthesized novel views. GenWarp produces unsatisfactory results due to the absence of depth information in its coordinate representation and the misalignment between its coordinate system and the input image. Due to space limitations, **more qualitative comparisons are provided in the Appendix**. It is worth noting that, unlike many video-based models listed here, our approach does not require generating intermediate frames between viewpoints, making it over an order of magnitude faster than video-based method to generate target view while still producing geometrically consistent results.

| Method | Tanks-and-Temples | | | RE10K | | | DL3DV | | |
|---|---|---|---|---|---|---|---|---|---|
| | PSNR↑ | SSIM↑ | LPIPS↓ | PSNR↑ | SSIM↑ | LPIPS↓ | PSNR↑ | SSIM↑ | LPIPS↓ |
| ZeroNVS (Sargent et al., 2024) | 13.14 | 0.327 | 0.516 | 15.23 | 0.540 | 0.386 | 14.17 | 0.441 | 0.481 |
| CameraCtrl (He et al., 2024) | 15.34 | 0.534 | 0.331 | 17.74 | 0.681 | 0.278 | 16.31 | 0.552 | 0.352 |
| GenWarp (Seo et al., 2024) | 16.45 | 0.513 | 0.377 | 15.30 | 0.538 | 0.371 | 15.81 | 0.531 | 0.382 |
| NVS-Solver (You et al., 2024) | 16.73 | 0.521 | 0.323 | 17.00 | 0.673 | 0.314 | 16.86 | 0.543 | 0.341 |
| ViewCrafter (Yu et al., 2024) | 17.18 | 0.589 | 0.346 | 17.75 | 0.681 | 0.315 | 17.24 | 0.571 | 0.329 |
| DimensionX (Sun et al., 2024) | 17.78 | 0.635 | 0.228 | 18.21 | 0.717 | 0.307 | 18.22 | 0.653 | 0.201 |
| SEVA (Zhou et al., 2025) | 17.61 | 0.621 | 0.235 | 17.58 | 0.688 | 0.334 | 18.01 | 0.638 | 0.214 |
| MVGenMaster (Cao et al., 2025) | 18.03 | 0.622 | 0.253 | 17.87 | 0.701 | 0.321 | 17.71 | 0.586 | 0.277 |
| See3D (Ma et al., 2025) | 18.35 | 0.641 | 0.244 | 18.24 | 0.735 | 0.293 | 18.41 | 0.631 | 0.215 |
| Voyager (Huang et al., 2025) | 18.61 | 0.669 | 0.238 | 18.56 | 0.723 | 0.264 | 18.84 | 0.636 | 0.227 |
| FlexWorld (Chen et al., 2025) | 18.91 | 0.675 | 0.236 | 18.03 | 0.691 | 0.282 | 18.67 | 0.645 | 0.218 |
| GEN3C (Ren et al., 2025) | 19.18 | 0.681 | 0.207 | 20.64 | 0.754 | 0.229 | 19.14 | 0.658 | 0.198 |
| Original PE | 20.03 | 0.683 | 0.221 | 20.17 | 0.752 | 0.233 | 19.92 | 0.667 | 0.201 |
| w/o Depth | 20.63 | 0.692 | 0.217 | 20.33 | 0.767 | 0.227 | 20.46 | 0.695 | 0.194 |
| w/o Multi-Level | 21.97 | 0.718 | 0.180 | 21.42 | 0.809 | 0.168 | 21.91 | 0.733 | 0.162 |
| **Ours** | **22.12** | **0.732** | **0.174** | **21.65** | **0.816** | **0.162** | **22.23** | **0.742** | **0.154** |

Table 1: Quantitative comparison of different methods on Tanks-and-Temples, RE10K, and DL3DV datasets. We report the average PSNR, SSIM, and LPIPS scores for novel view synthesis from a single input image.

Beyond pose-conditioned approaches, recent image editing models such as Flux.1 Kontext (Labs et al., 2025) and Qwen-Image-Edit (Wu et al., 2025a) also demonstrate strong capabilities in viewpoint manipulation. We further compare our method with these prompt-based editing results, as illustrated in Figure 6. Flux is generally insensitive to prompts specifying spatial viewpoint changes, often producing only minor viewpoint variations while introducing noticeable artifacts. Qwen, on the other hand, achieves more pronounced spatial editing effects than Flux, but tends to alter the original image tokens. As shown in the rightmost example of Figure 6, the result appears overly smoothed and even alters the person's identity. Overall, it remains very challenging to precisely control viewpoint changes through prompts. More comparisons can be found in the **Appendix**.

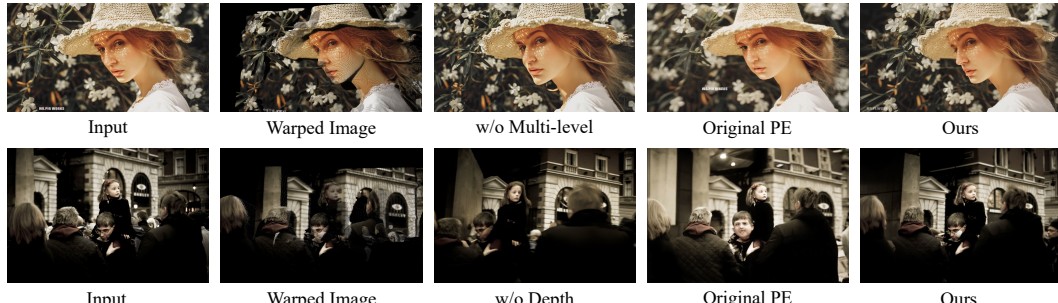

| Input | Warped Image | w/o Multi-level | Original PE | Ours |
|---|---|---|---|---|
| Input | Warped Image | w/o Depth | Original PE | Ours |

Figure 7: Ablation studies. Removing the detailed positional encoding or depth leads to different types of degradation in the generated results.

## 4.3 ABLATION STUDIES

We mainly analyze the effect of removing our two key components: the hierarchical detailed positional encodings and the additional depth-aware extension. The quantitative impact of removing each component can be observed in Table 1, while Figure 7 provides two illustrative cases. As shown in the top example of Figure 7, when the multi-level positional encoding (particularly the detailed level) is removed, undesirable distortions appear due to the mismatch between patch-level positional encodings and the reconstruction. When depth information is removed (see bottom example in Figure 7), the generated images suffer from severe spatial misalignment.

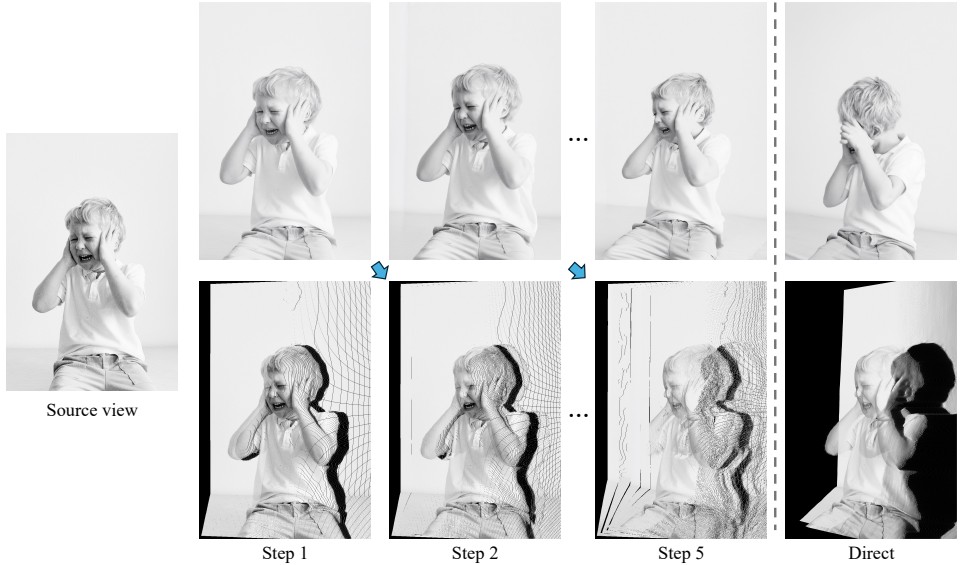

Figure 8: Multi-step generation. Left: input image. Top: generated results. Bottom: rotated point clouds. Right: direct one-step generation.

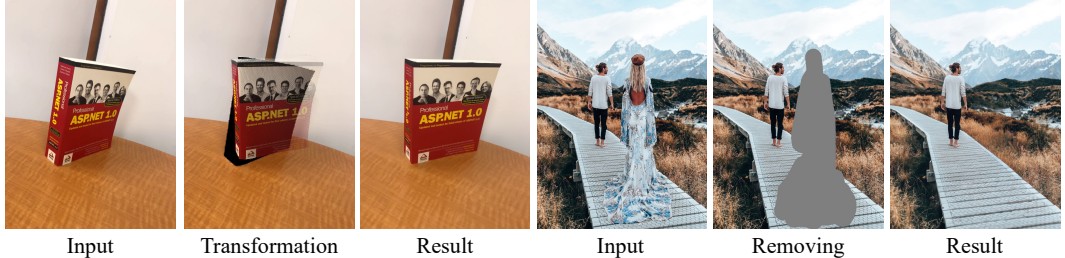

Figure 9: Applications. The left example shows object 3D editing, while the right example shows object removal, highlighting the versatility of our model in different spatial editing tasks.

When applying our method to generate results under large viewpoint changes, the model is required to directly generate a substantial amount of unseen content, which increases the generation burden and may compromise consistency with the source image. To mitigate this issue, we decompose the transformation into multiple steps, in which the model only needs to complete a small portion of the missing content in each step. As shown in Figure 8, we divide the transformation of the target viewpoint into five steps. After each step, the newly generated content is fused back into the image tokens of the original viewpoint, and the fused tokens (or point cloud) are then transformed to the next intermediate viewpoint for subsequent generation. Compared to directly transforming to the target viewpoint in one step (rightmost result in Figure 8), this progressive strategy produces results that are more consistent with the source view. See **Appendix** for **quantitative** comparisons.

## 4.4    OTHER APPLICATIONS

After training, our NVS model acquires the ability to reason over visual tokens in 3D space and generate consistent content. Consequently, it can naturally adapt to other tasks with similar spatial logic, even in the **absence of task-specific training**. As illustrated in Figure 9, in the left example we perform object-level 3D editing by isolating the point cloud of the book, rotating it to a new viewpoint, and recomposing it with the original background. In the right example, we achieve object removal by discarding the tokens corresponding to the masked human region and replenishing them with noise, resulting in a realistic removal effect. More results can be found in the **Appendix**.

## 5 CONCLUSIONS

In this work, we revisited the internal mechanisms of Diffusion Transformers and revealed that spatial coherence is largely governed by positional encodings rather than explicit token interactions. Building on this observation, we introduced the Positional Encoding Field (PE-Field), which extends standard 2D encodings into a 3D, depth-aware and hierarchical framework. This design equips DiTs with geometry-aware generative capabilities, achieving state-of-the-art results on single-image novel view synthesis while also enabling flexible and controllable spatial image editing. We hope our study sheds light on the overlooked role of positional encodings and inspires future research into more principled and spatially grounded generative architectures.

**Acknowledgements.** This work was supported by NSF-2047677, NSF-2413161, NSF-2504906, NSF-2515626, and GIFTs from Adobe and Google. This work was supported by computing support on the Vista GPU Cluster through the Center for Generative AI (CGAI) and the Texas Advanced Computing Center (TACC) at UT Austin.

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
