# POSITIONAL ENCODING FIELD
# −SUPPLEMENTARY MATERIALS−

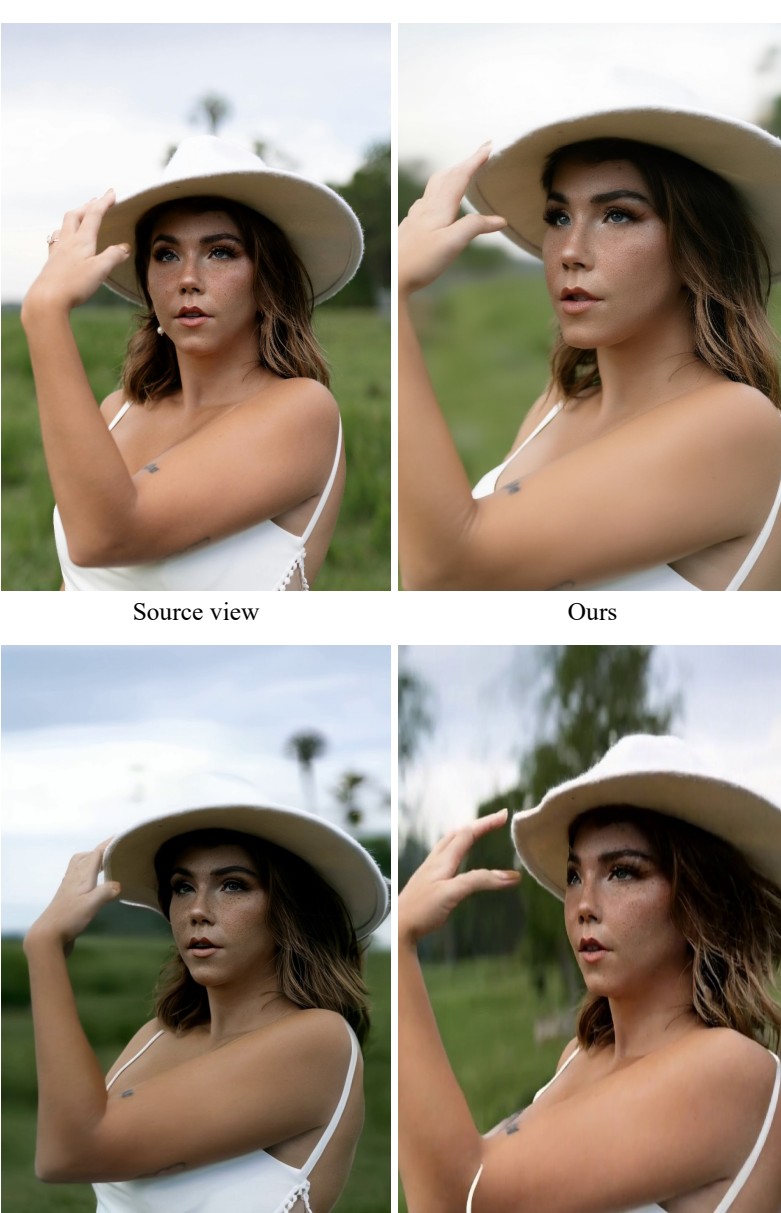

Source view           Ours

ViewCrafter           Gen3C

Figure 1: Supplementary novel view synthesis (NVS) examples on in-the-wild images. For the 30°
rightward rotation, ViewCrafter produces only a small rotation with hand distortions and skin color
changes, GEN3C introduces facial distortions.

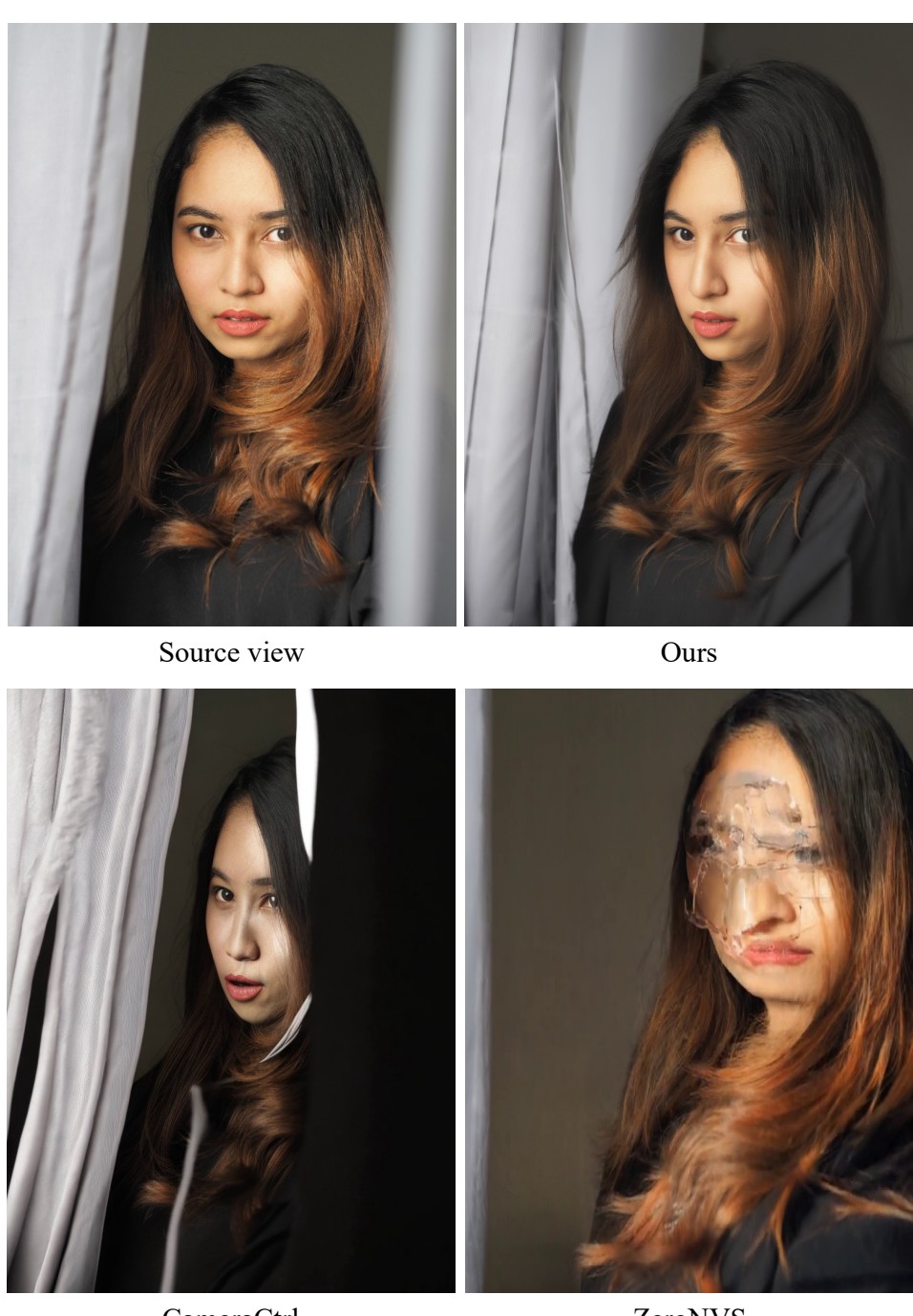

Figure 2: Supplementary novel view synthesis (NVS) examples on in-the-wild images. For the 30°
rightward rotation, CameraCtrl and ZeroNVS both cause facial distortions.

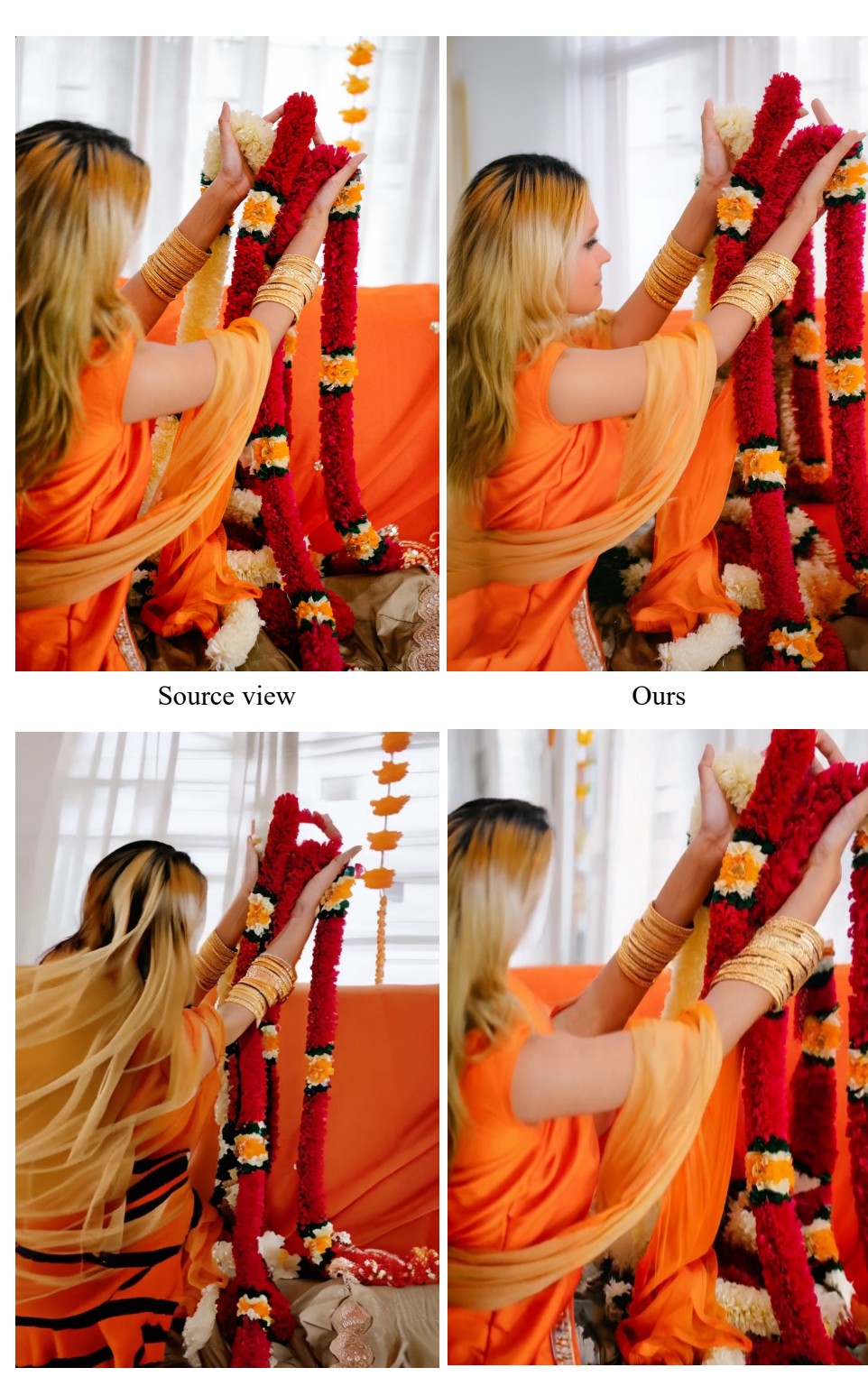

Source view       Ours

See3D       MVGenMaster

Figure 3: Supplementary novel view synthesis (NVS) examples on in-the-wild images. For the 30°
rightward rotation, See3D yields inaccurate viewpoint changes and significantly alters the original
appearance, MVGenMaster fails to complete reasonable content for the human face.

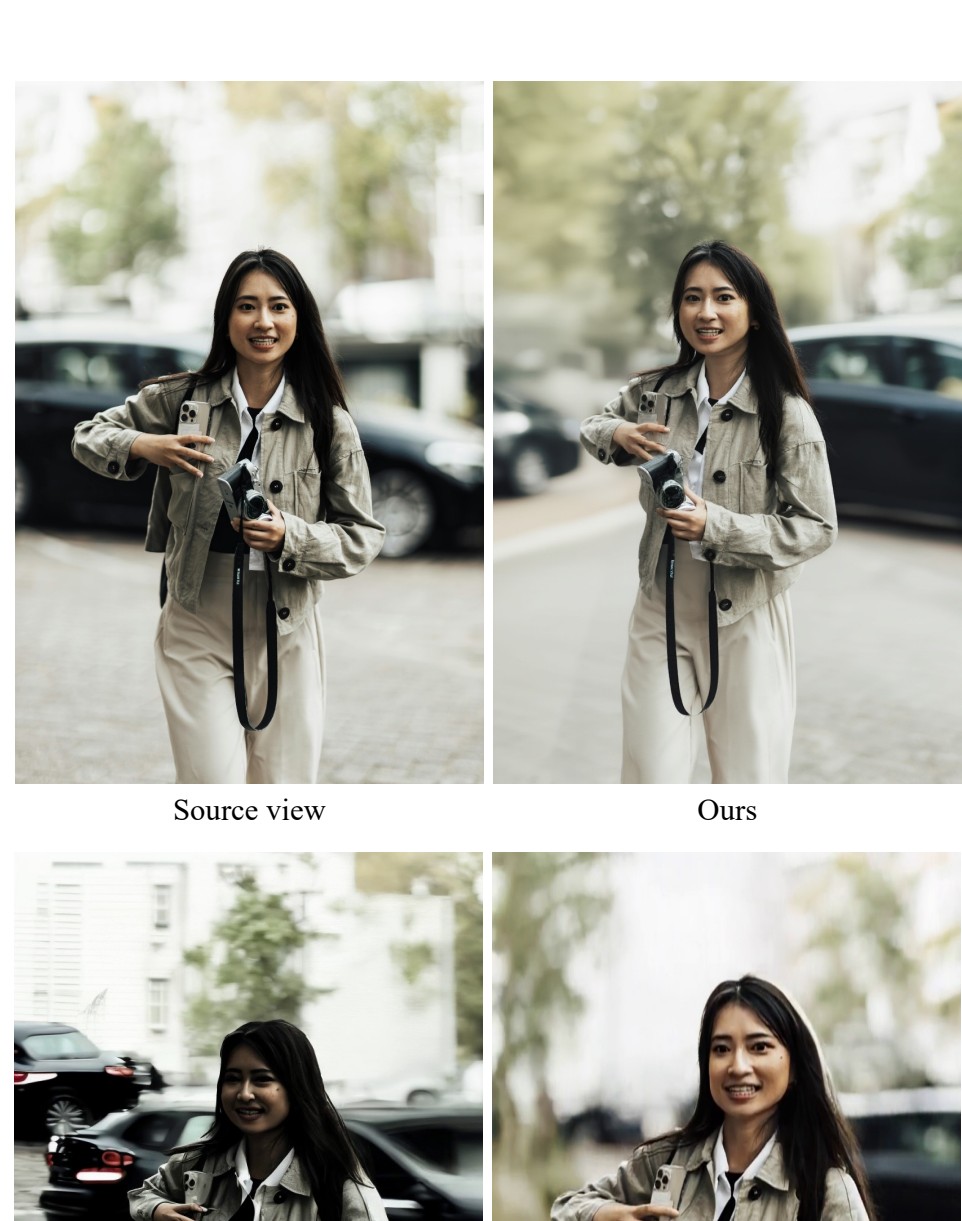

Source view                         Ours

ViewCrafter                         Gen3C

Figure 4: Supplementary novel view synthesis (NVS) examples on in-the-wild images. For the 30° rightward rotation, ViewCrafter produces only a small rotation with hand distortions and skin color changes, GEN3C introduces facial distortions.

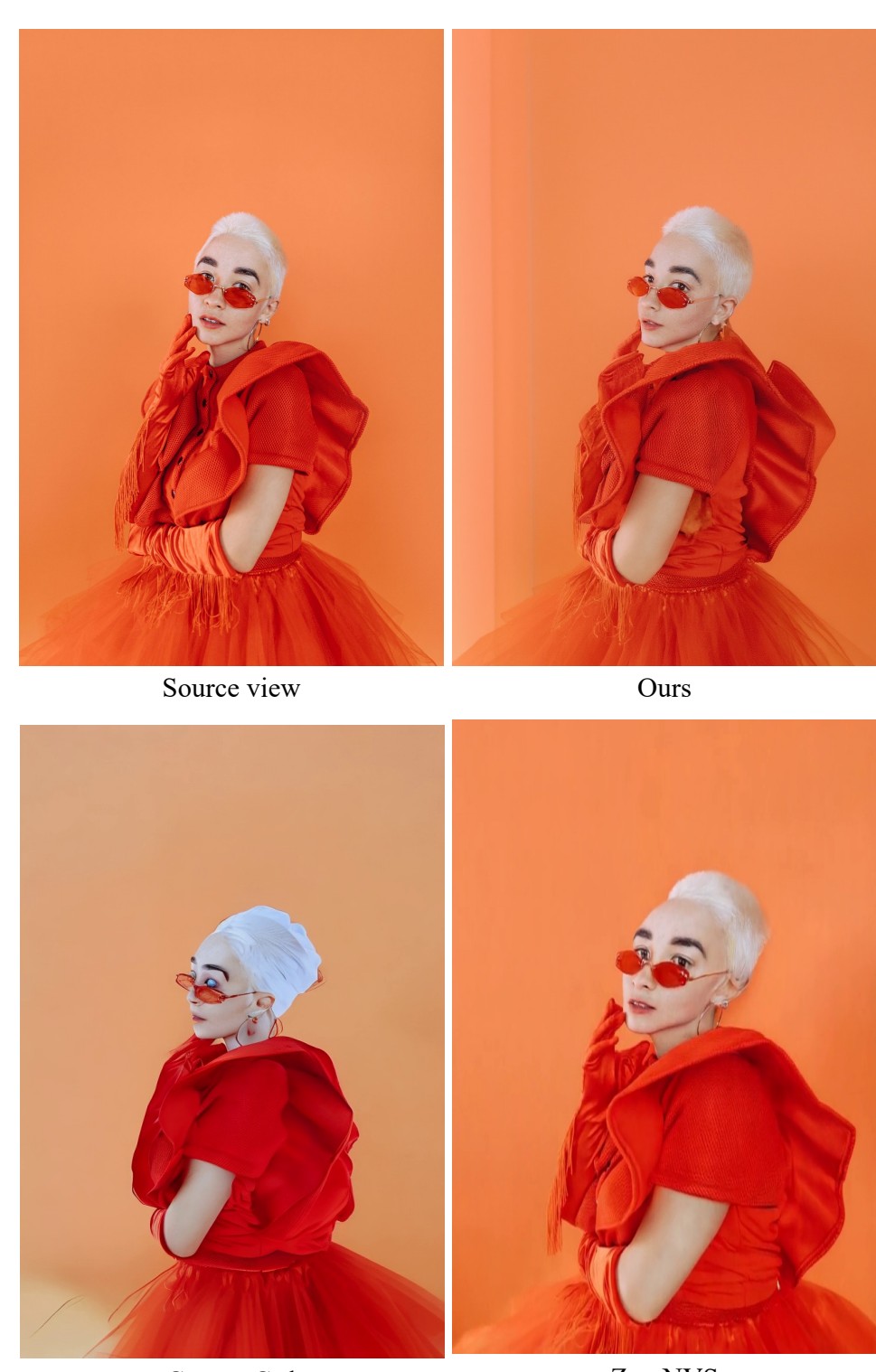

Figure 5: Supplementary novel view synthesis (NVS) examples on in-the-wild images. For the 30°
rightward rotation, CameraCtrl and ZeroNVS both cause facial distortions.

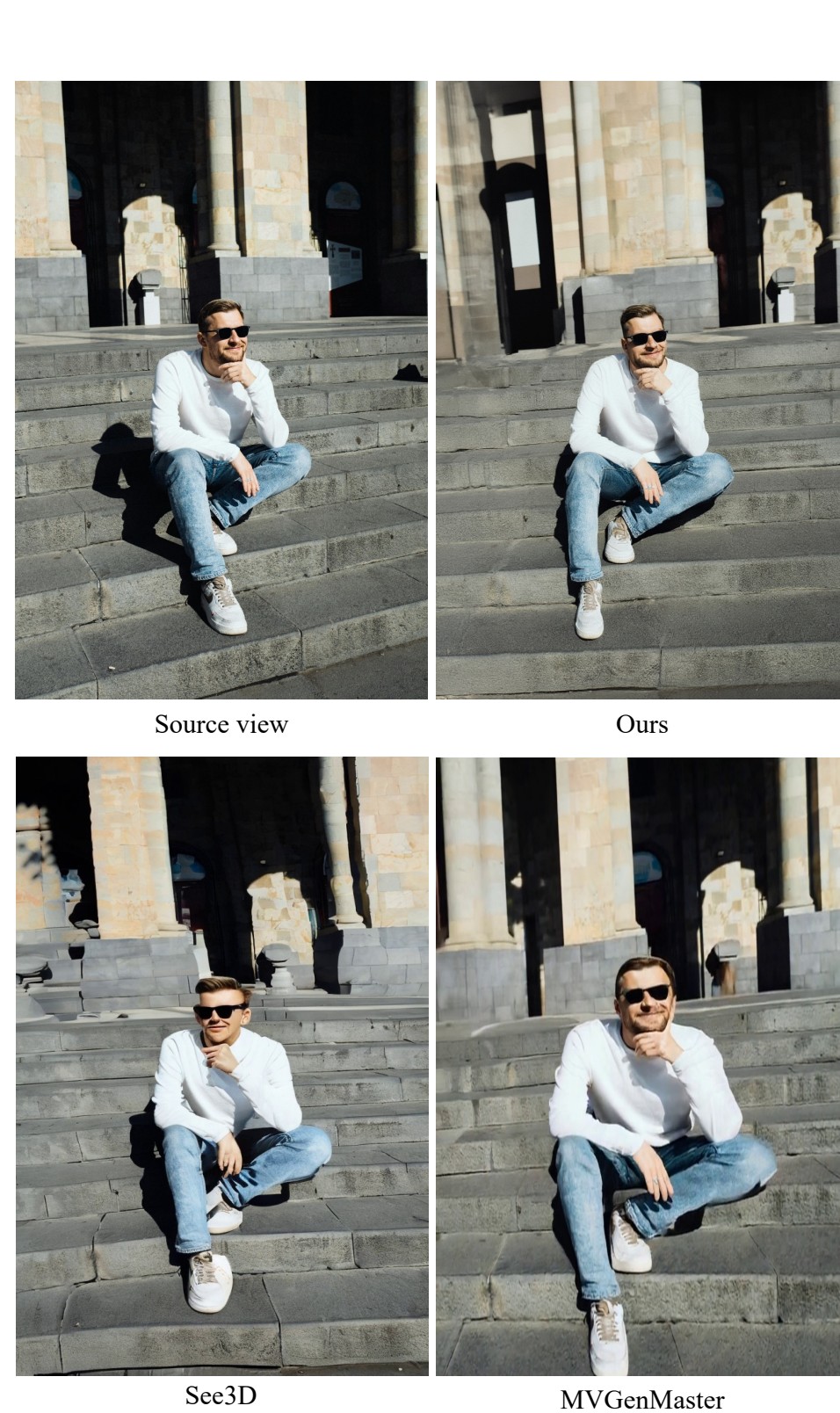

Figure 6: Supplementary novel view synthesis (NVS) examples on in-the-wild images. For the 30° rightward rotation, See3D and MVGenMaster both introduce distortions in the human face and shoes.

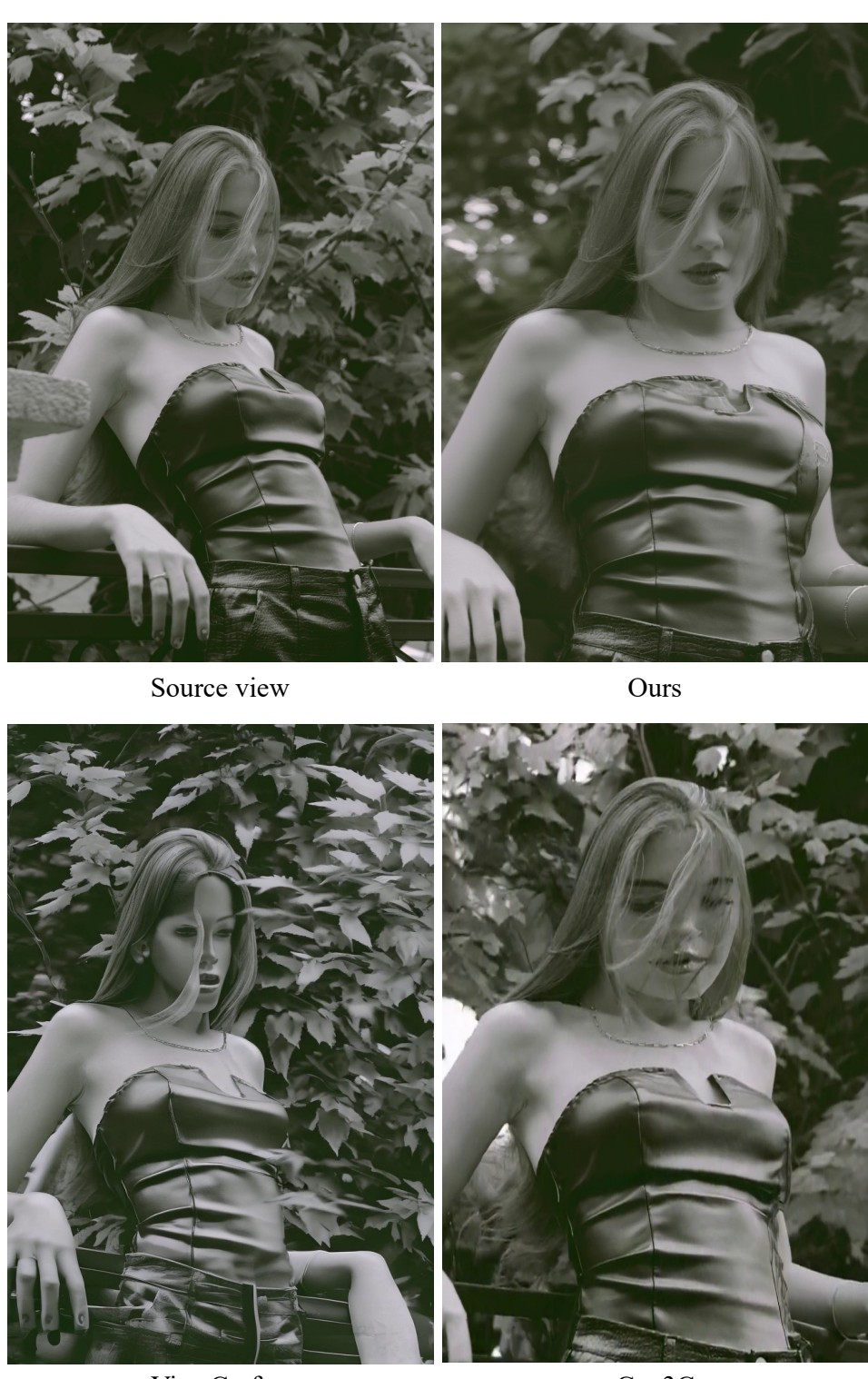

Source view

Ours

ViewCrafter

Gen3C

Figure 7: Supplementary novel view synthesis (NVS) examples on in-the-wild images. For the 30° rightward rotation, ViewCrafter produces only a small rotation but completely distorts the face, GEN3C introduces facial distortions.

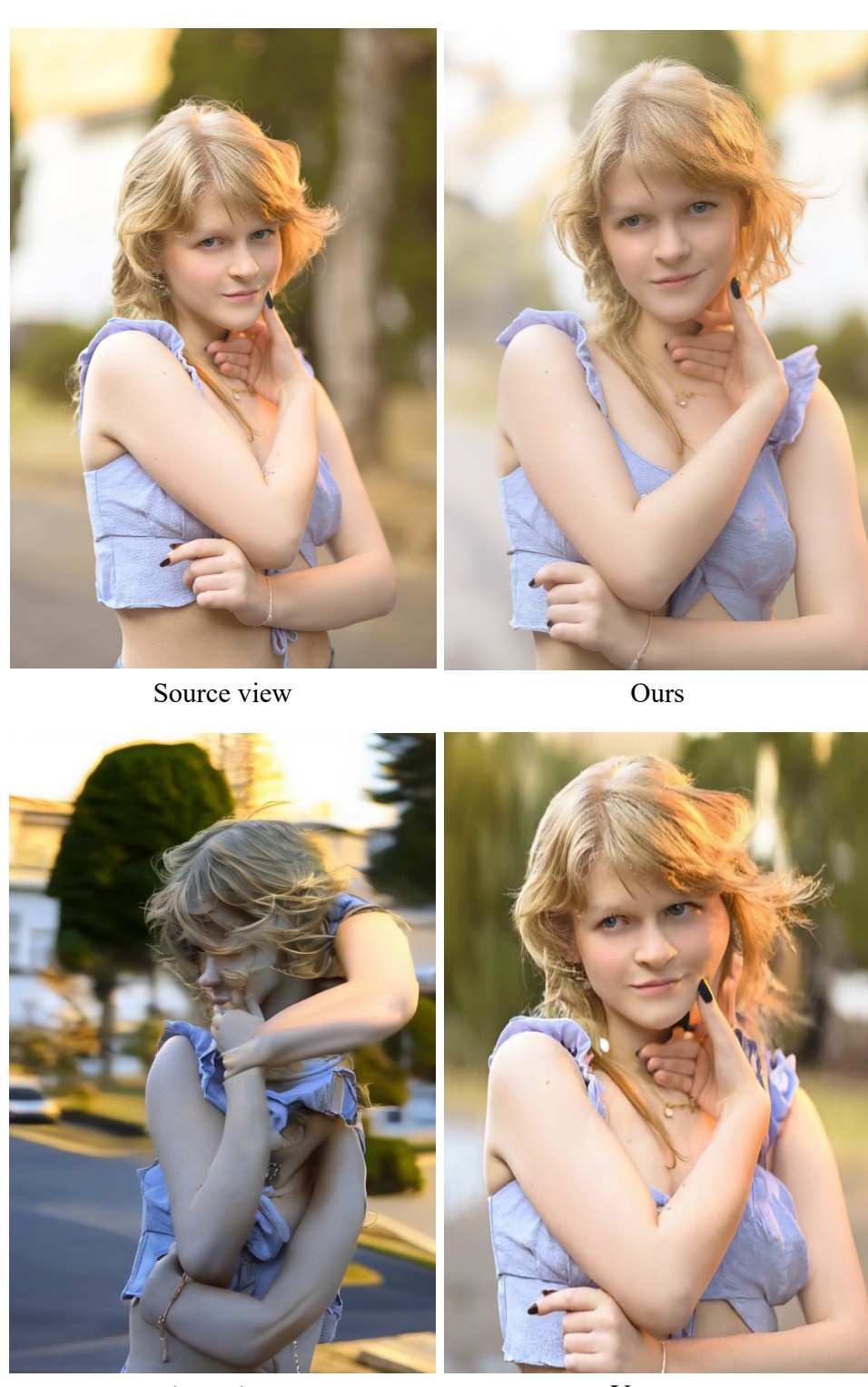

Figure 8: Supplementary novel view synthesis (NVS) examples on in-the-wild images. For the 30°
rightward rotation, DimensionX completely distorts the human figure, Voyager introduces facial
distortions.

Figure 9: Supplementary novel view synthesis (NVS) examples on in-the-wild images. For the 30°
rightward rotation, SEVA completely distorts the human face, FlexWorld introduces facial distor-
tions.

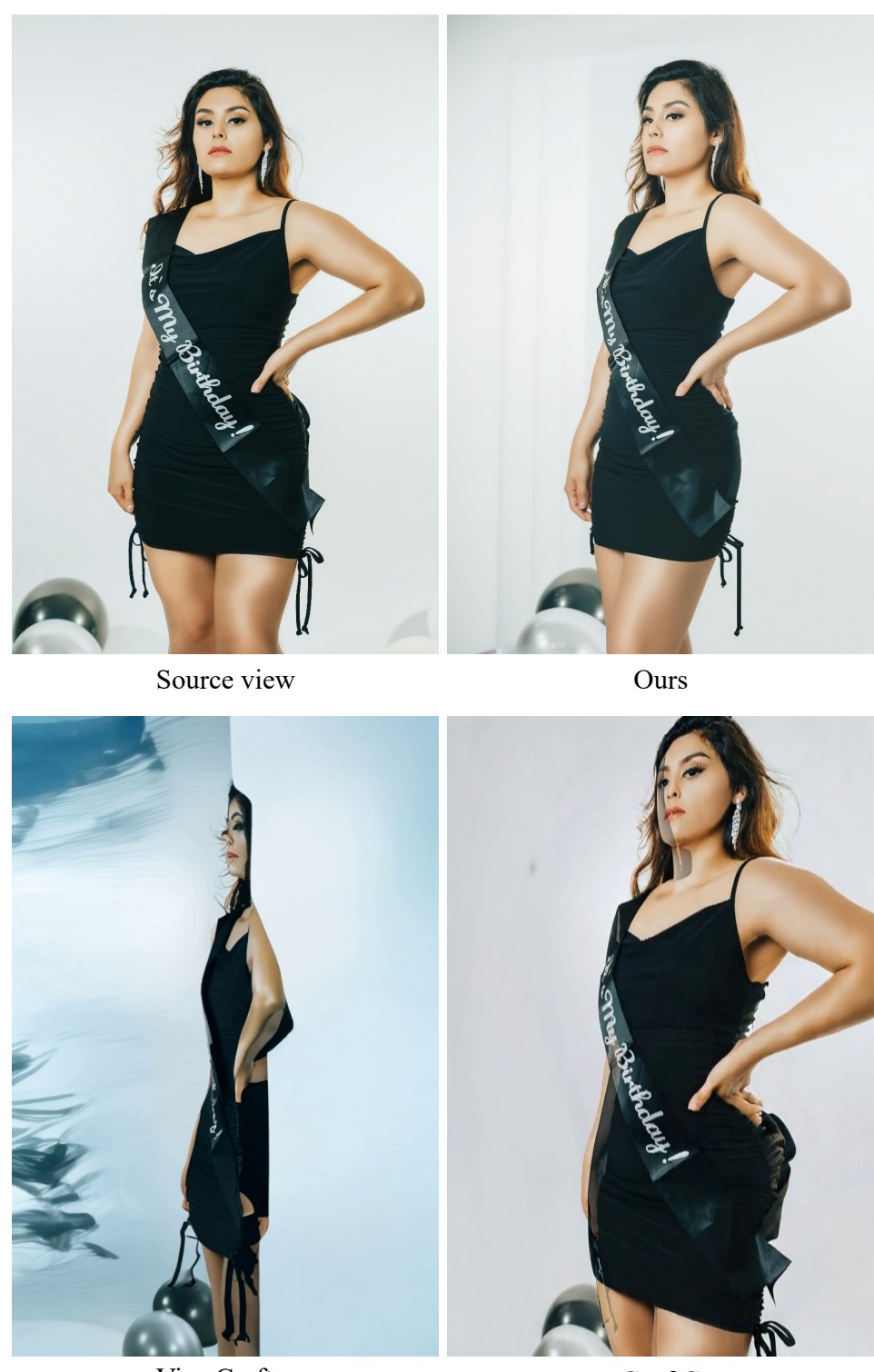

Source view          Ours

ViewCrafter          Gen3C

Figure 10: Supplementary novel view synthesis (NVS) examples on in-the-wild images. For the 30° rightward rotation, ViewCrafter completely distorts the human figure, GEN3C introduces facial artifacts.

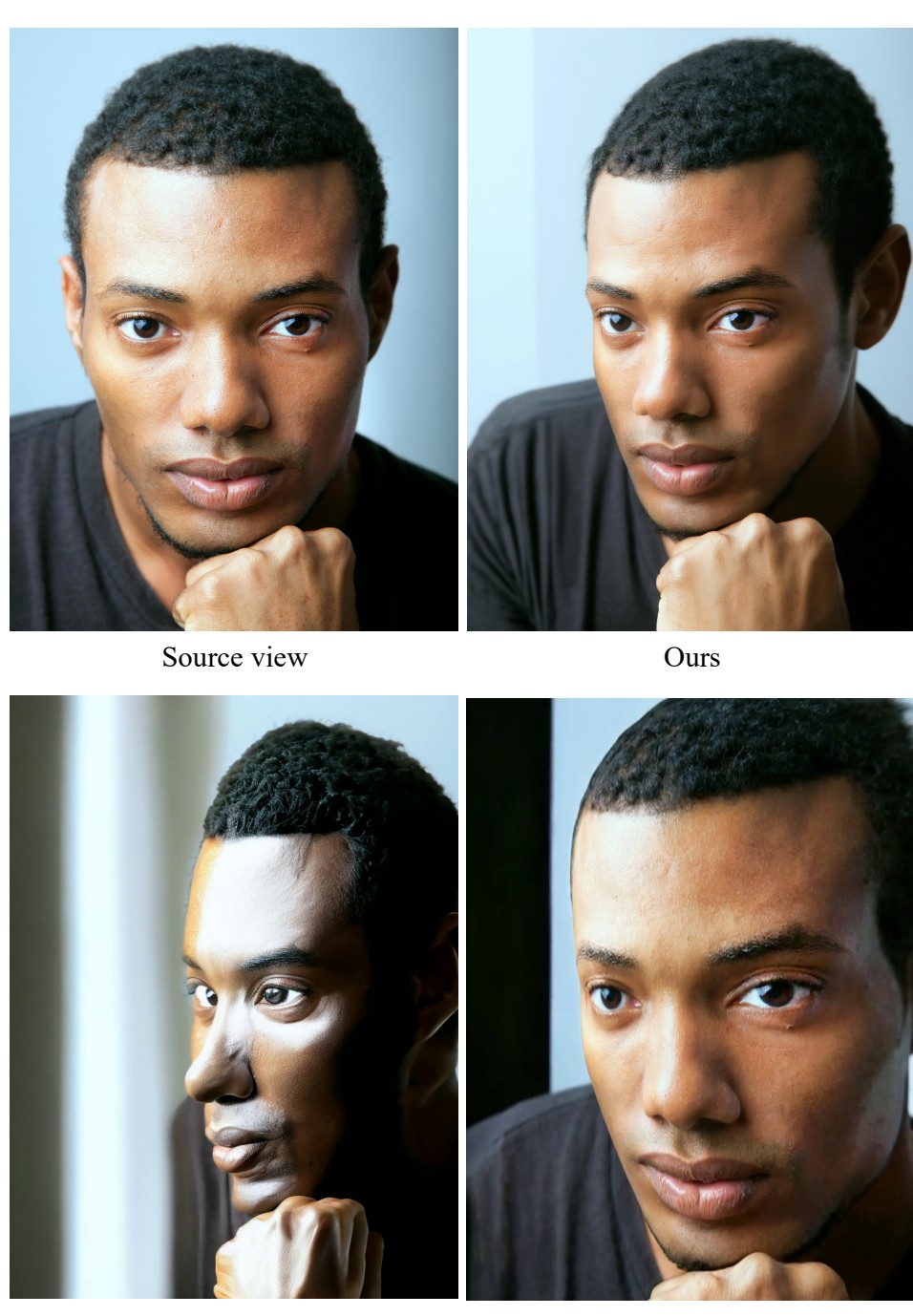

Figure 11: Supplementary novel view synthesis (NVS) examples on in-the-wild images. For the 30°
rightward rotation, CameraCtrl and ZeroNVS both cause facial distortions.

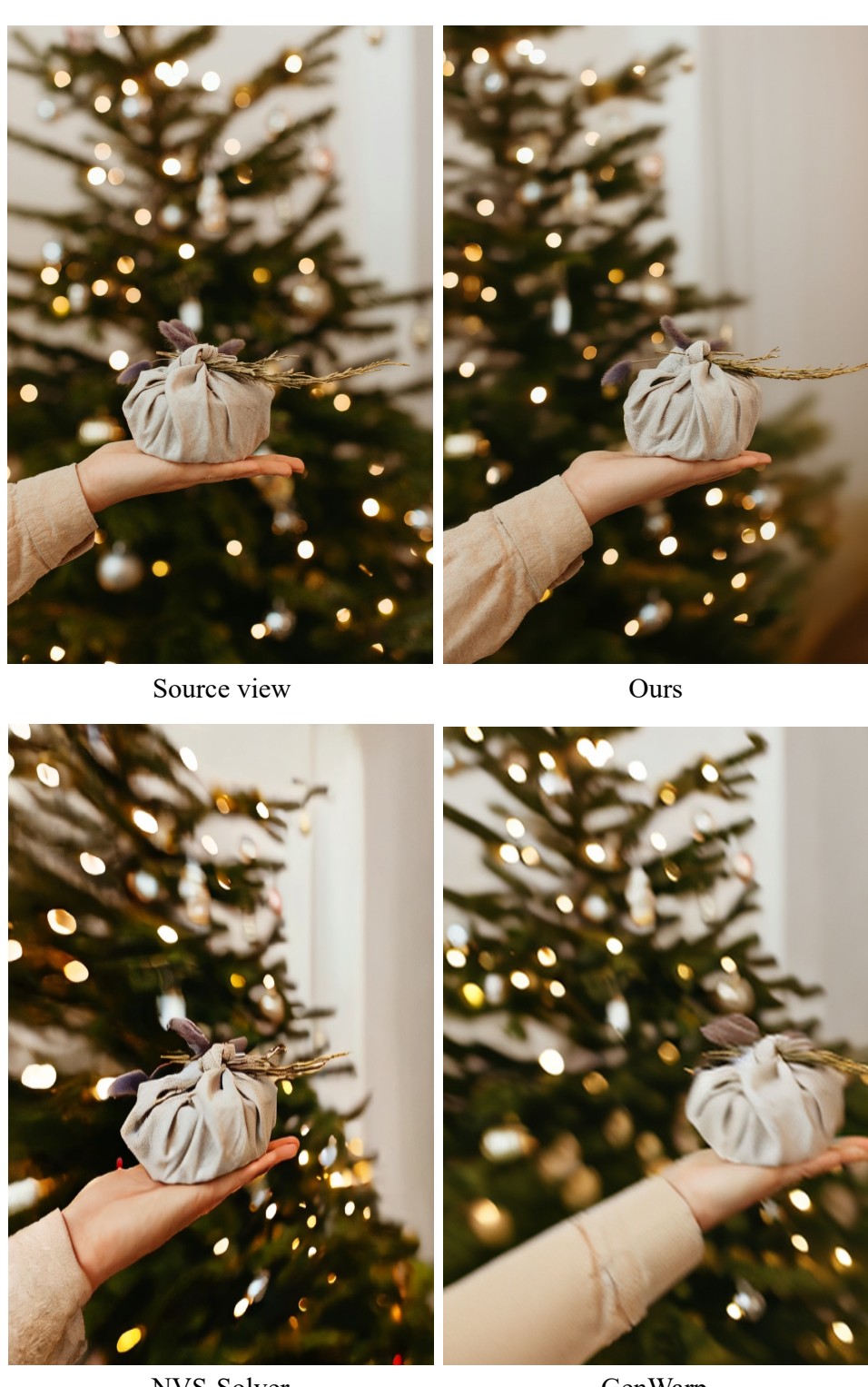

Figure 12: Supplementary novel view synthesis (NVS) examples on in-the-wild images. For the 30° leftward rotation, NVS-Solver shows little change but introduces some distortions, GenWarp produces blurred results.

648
649
650
651
652
653
654
655
656
657
658
659
660
661
662
663
664
665
666
667
668
669
670
671
672
673
674
675
676
677
678
679
680
681
682
683
684
685
686
687
688
689
690
691
692
693
694
695
696
697
698
699
700
701

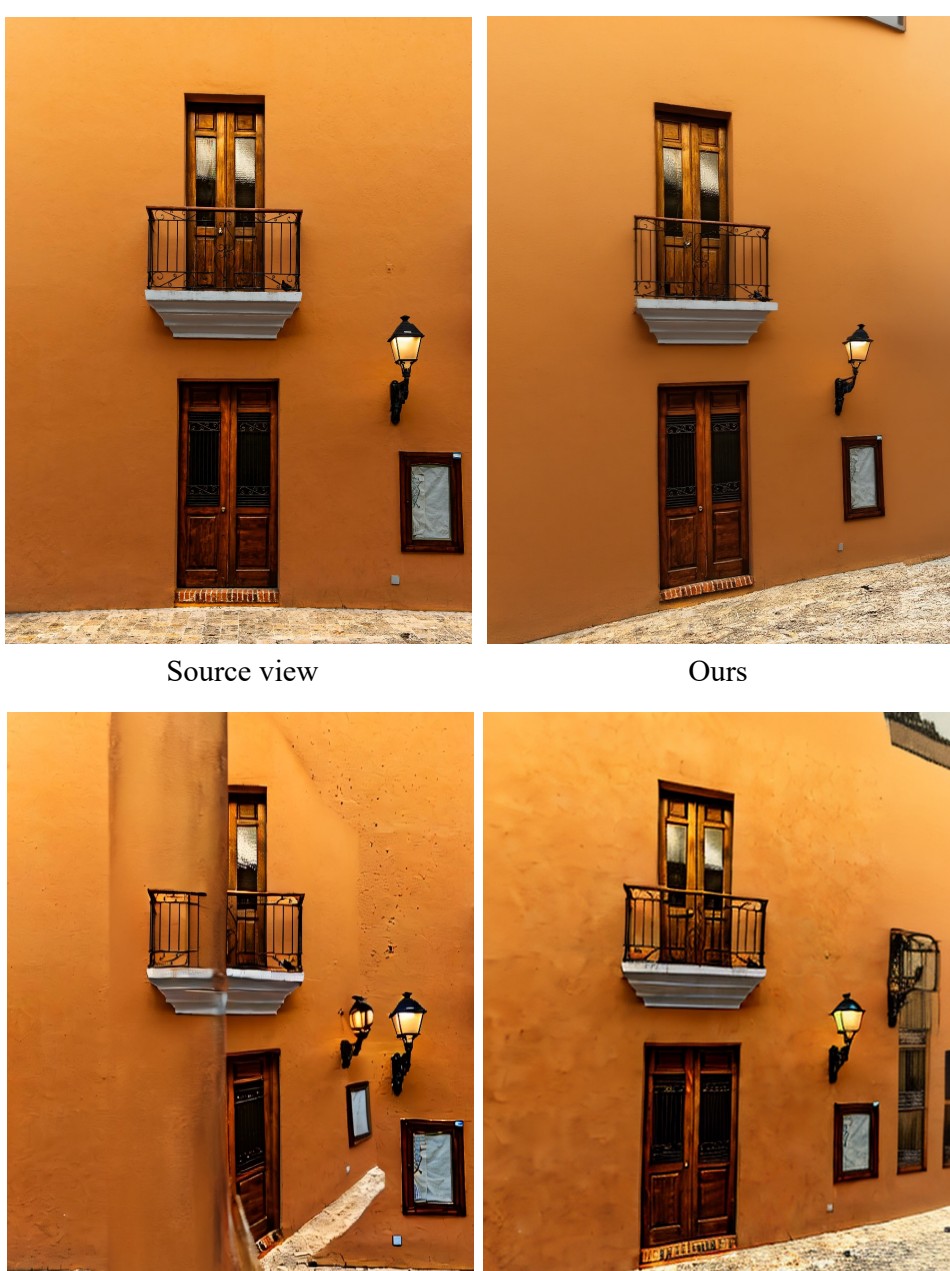

Figure 13: Supplementary novel view synthesis (NVS) examples on in-the-wild images. For the 30° leftward rotation, See3D distorts the scene, and MVGenMaster adds noise to the results.

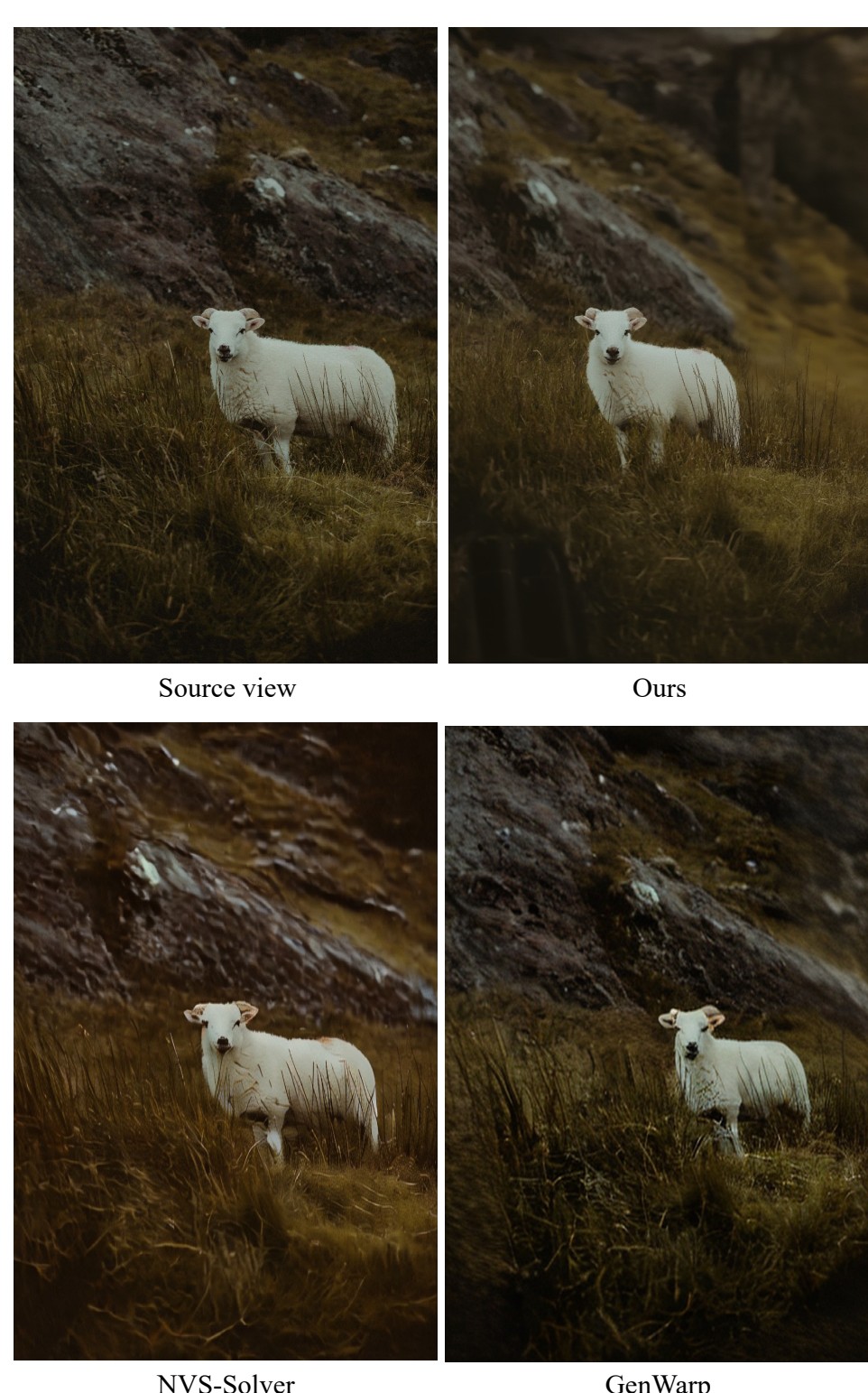

Source view          Ours

NVS-Solver          GenWarp

Figure 14: Supplementary novel view synthesis (NVS) examples on in-the-wild images. For the 30° leftward rotation, NVS-Solver introduces distortions on the sheep and alters the overall color tone, GenWarp produces smaller changes but still distorts the sheep.

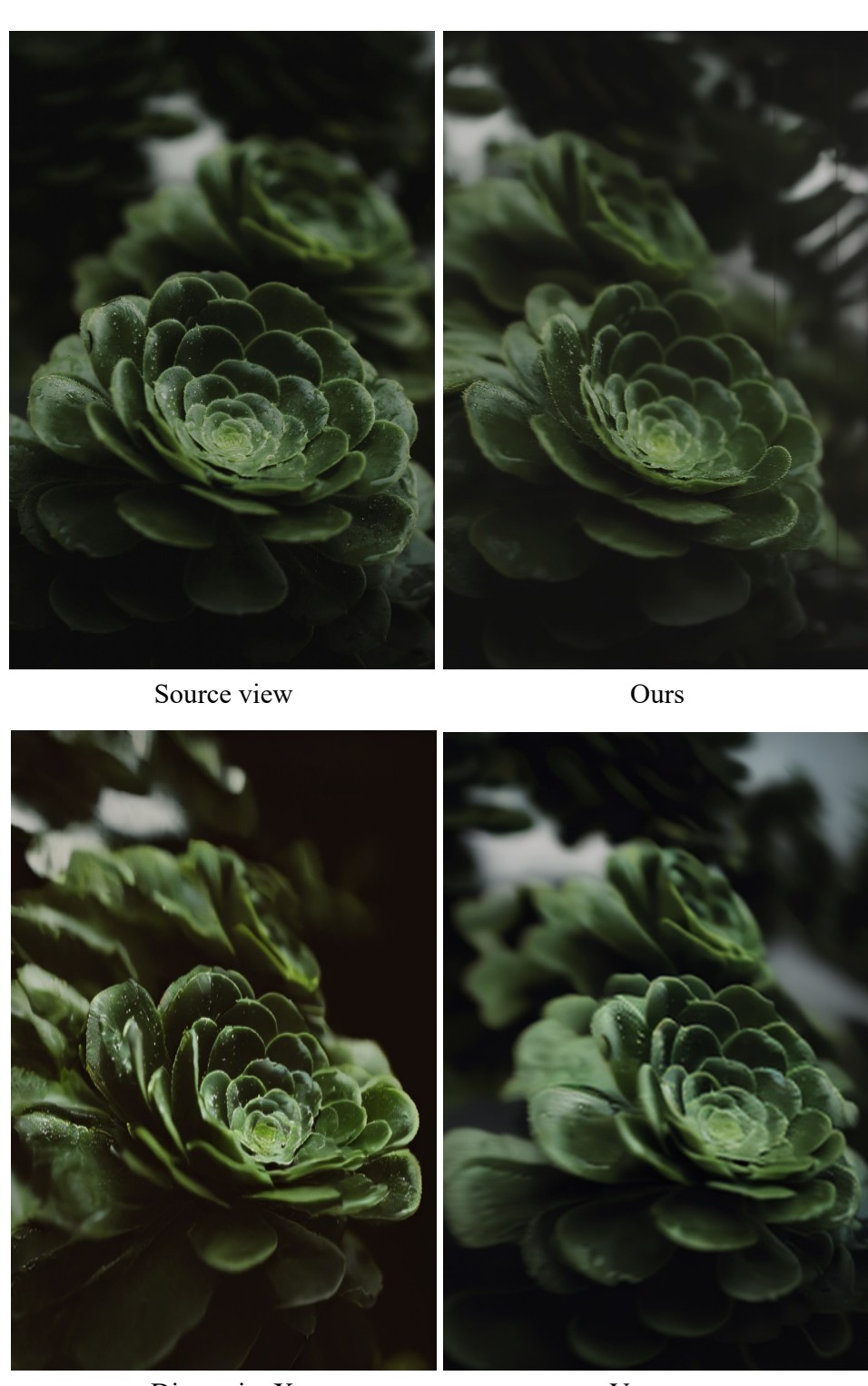

Source view — Ours

DimensionX — Voyager

Figure 15: Supplementary novel view synthesis (NVS) examples on in-the-wild images. For the 30° leftward rotation, DimensionX and Voyager both alter the shape of the flowers.

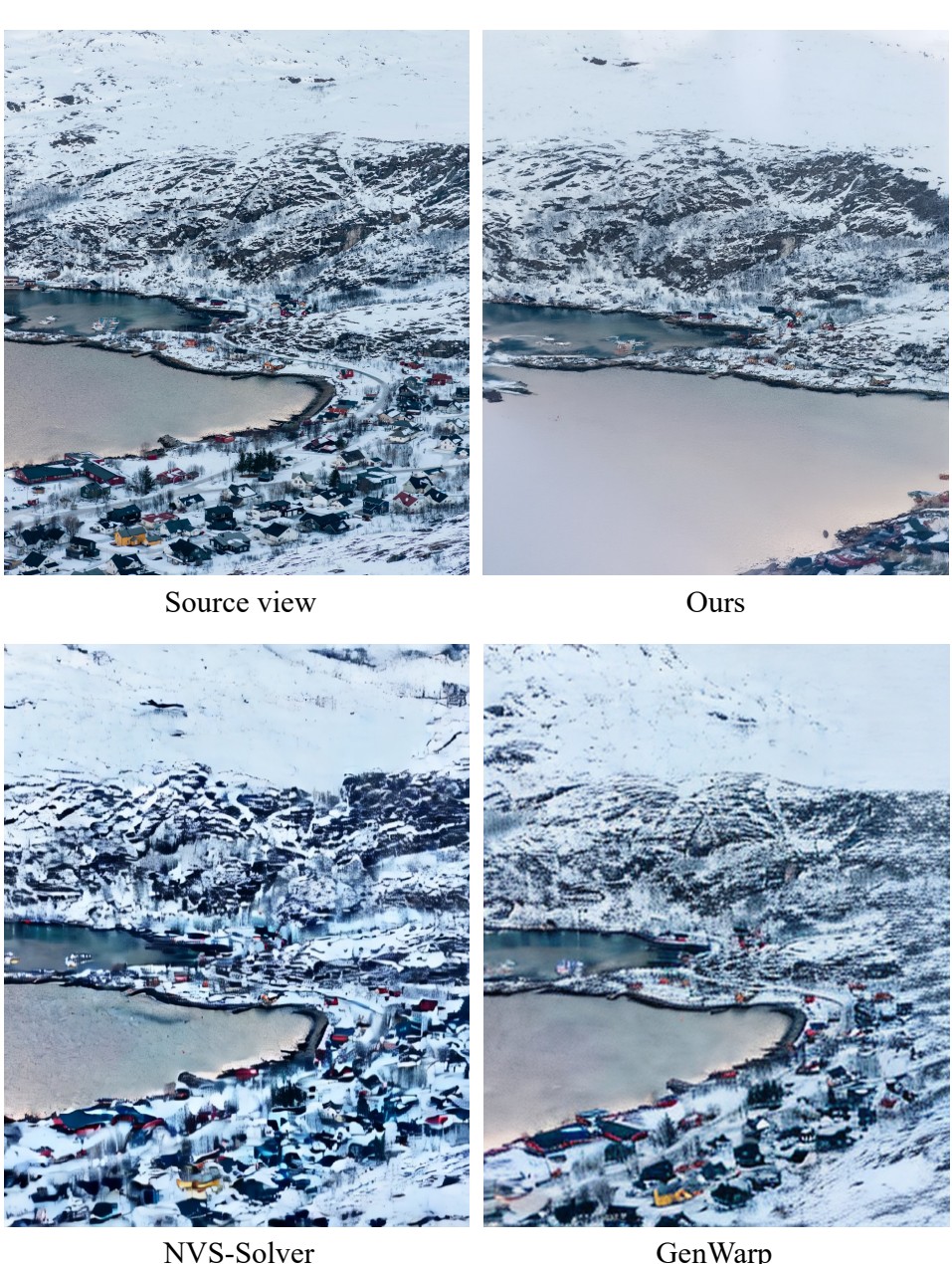

Source view             Ours

NVS-Solver          GenWarp

Figure 16: Supplementary novel view synthesis (NVS) examples on in-the-wild images. For the $30°$ leftward rotation, NVS-Solver and GenWarp show little change but generate blurred images.

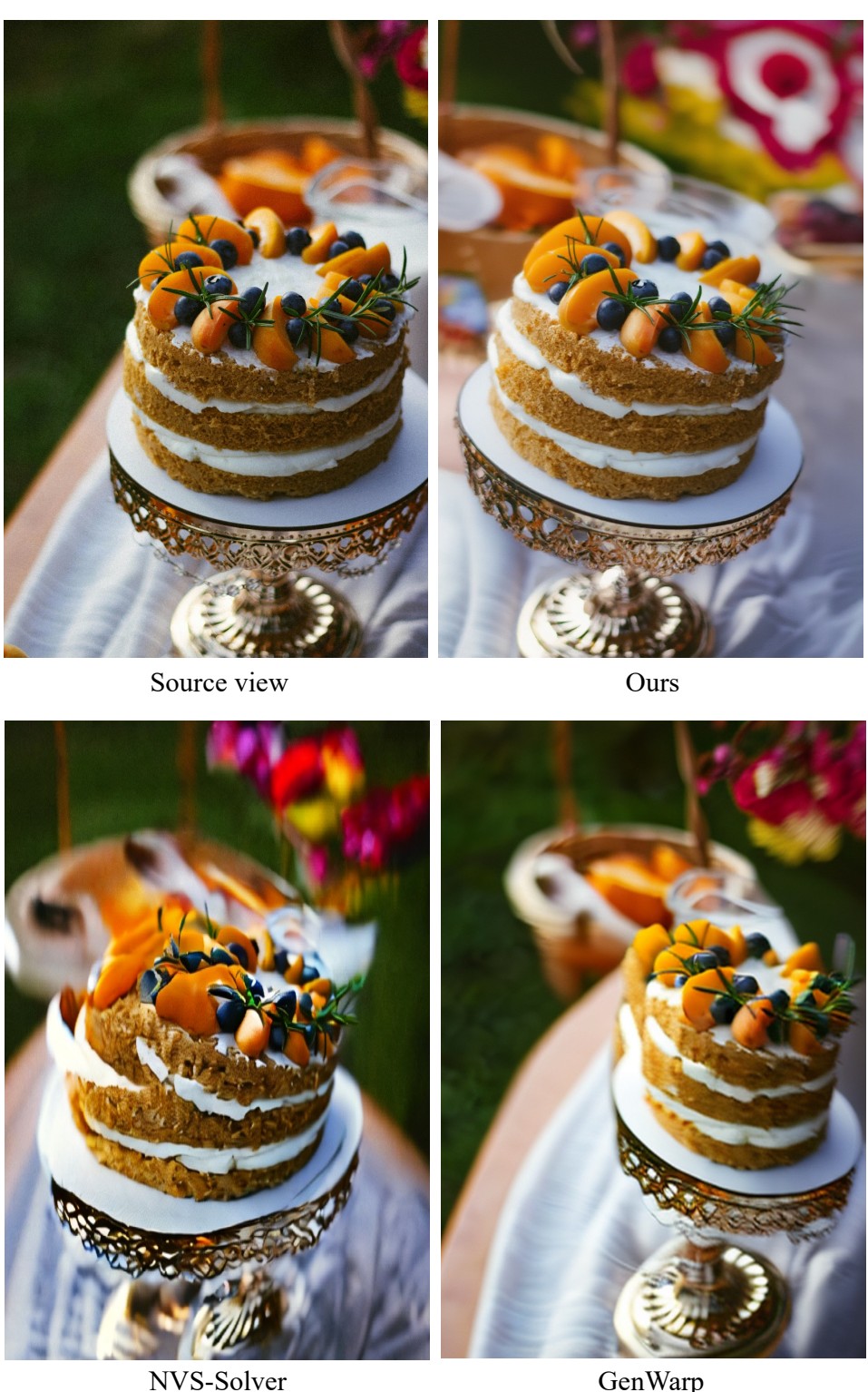

Figure 17: Supplementary novel view synthesis (NVS) examples on in-the-wild images. For the 30°
leftward rotation, NVS-Solver and GenWarp both introduce object distortions and produce blurred
results.

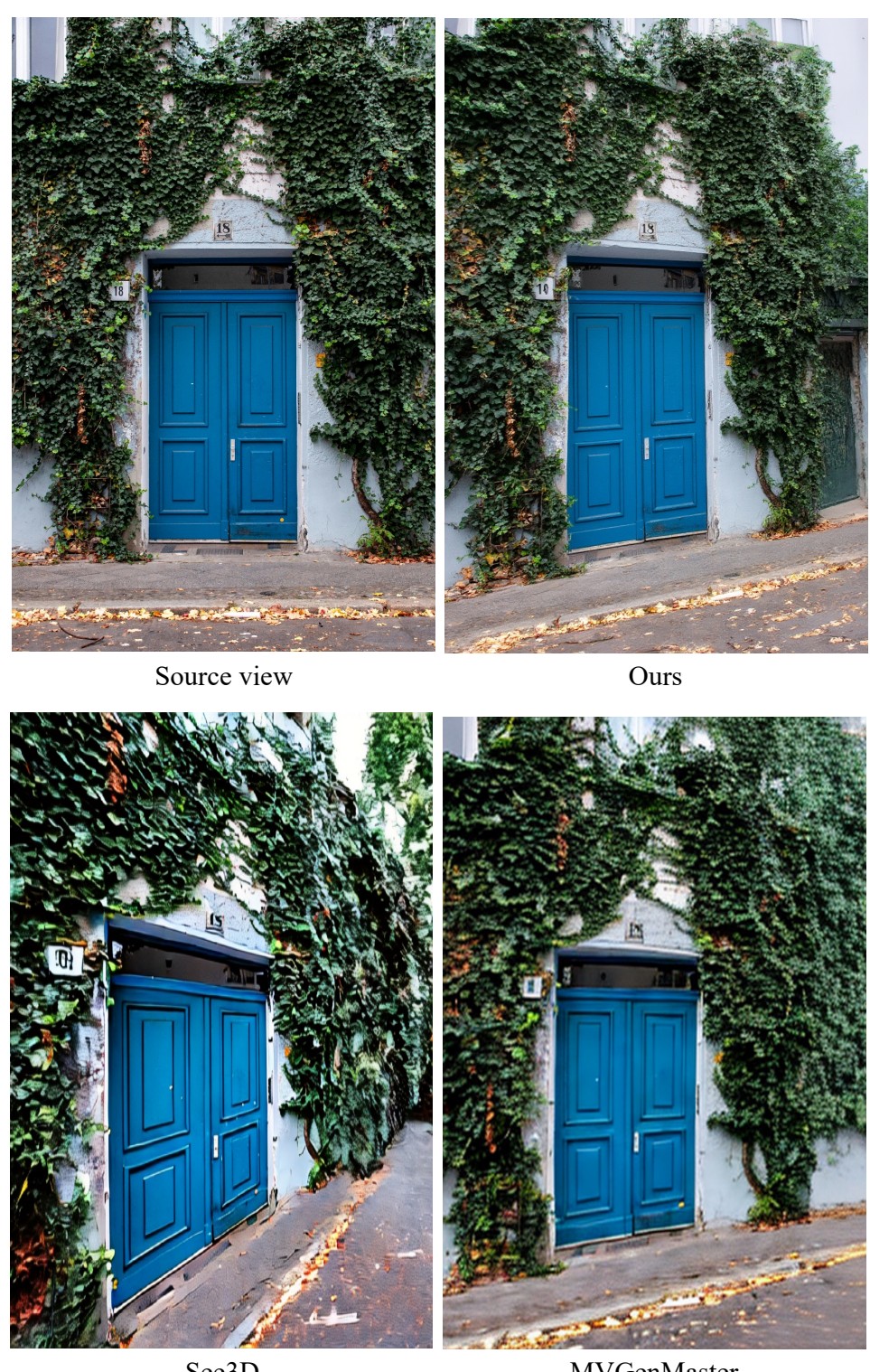

Figure 18: Supplementary novel view synthesis (NVS) examples on in-the-wild images. For the 30°
leftward rotation, See3D distorts the scene, and MVGenMaster adds noise to the results.

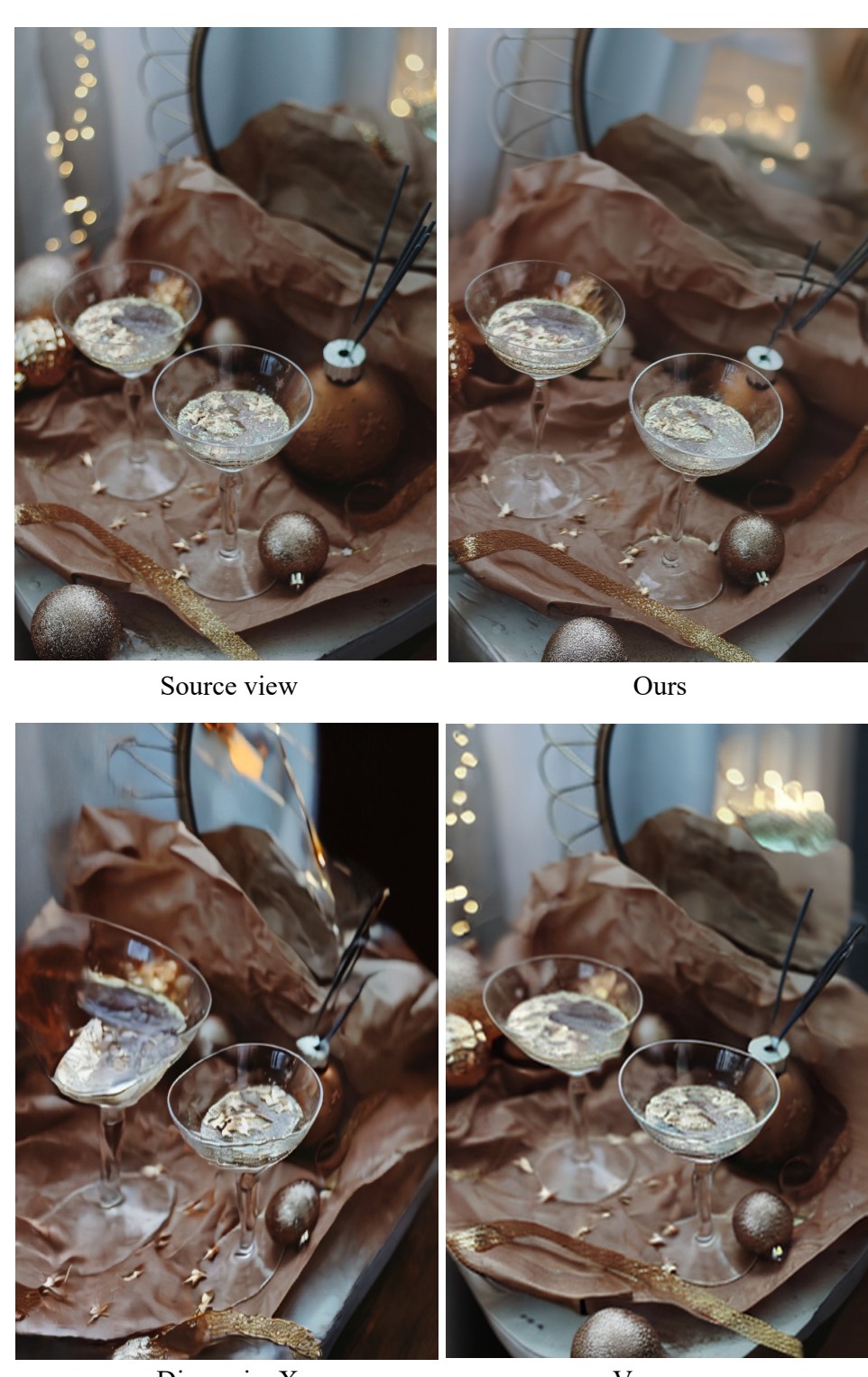

Figure 19: Supplementary novel view synthesis (NVS) examples on in-the-wild images. For the 30° leftward rotation, DimensionX and Voyager both alter the shape of the goblet.

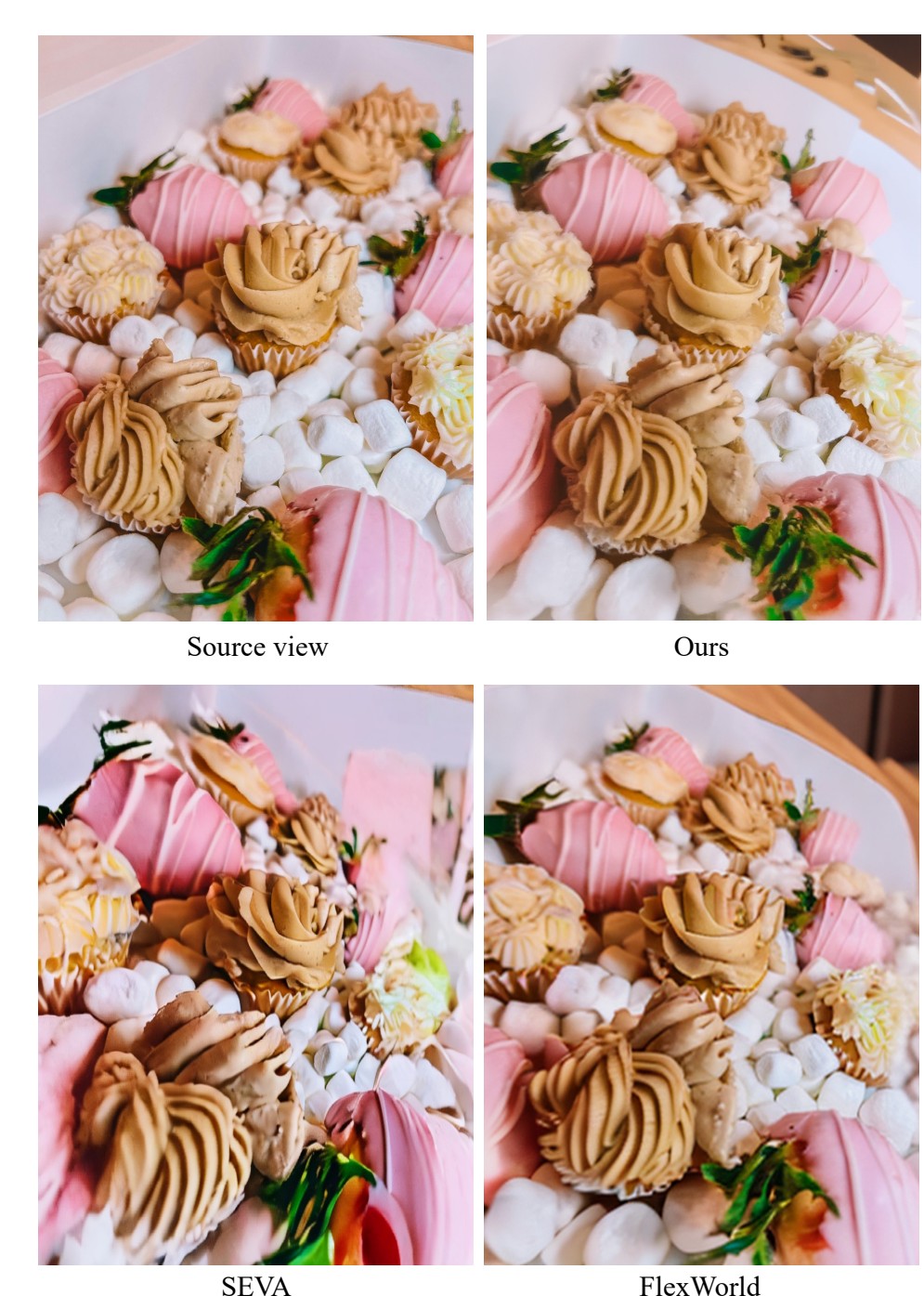

Figure 20: Supplementary novel view synthesis (NVS) examples on in-the-wild images. For the 30° leftward rotation, SEVA produces completely distorted results, FlexWorld introduces object distortions and alters the overall image style.

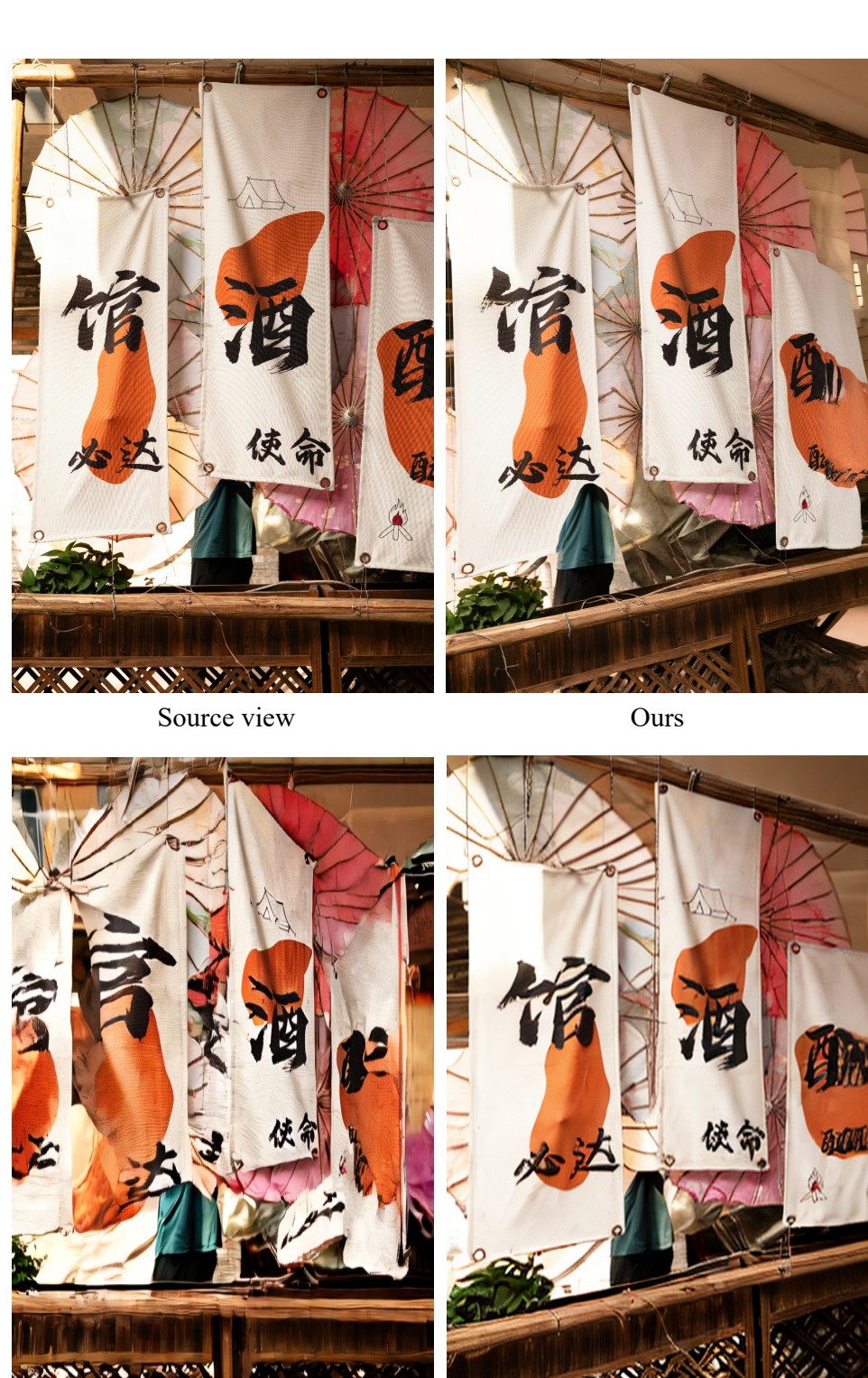

Source view             Ours

NVS-Solver             GenWarp

Figure 21: Supplementary novel view synthesis (NVS) examples on in-the-wild images. For the 30° leftward rotation, NVS-Solver shows little change but introduces some distortions, GenWarp produces blurred results.

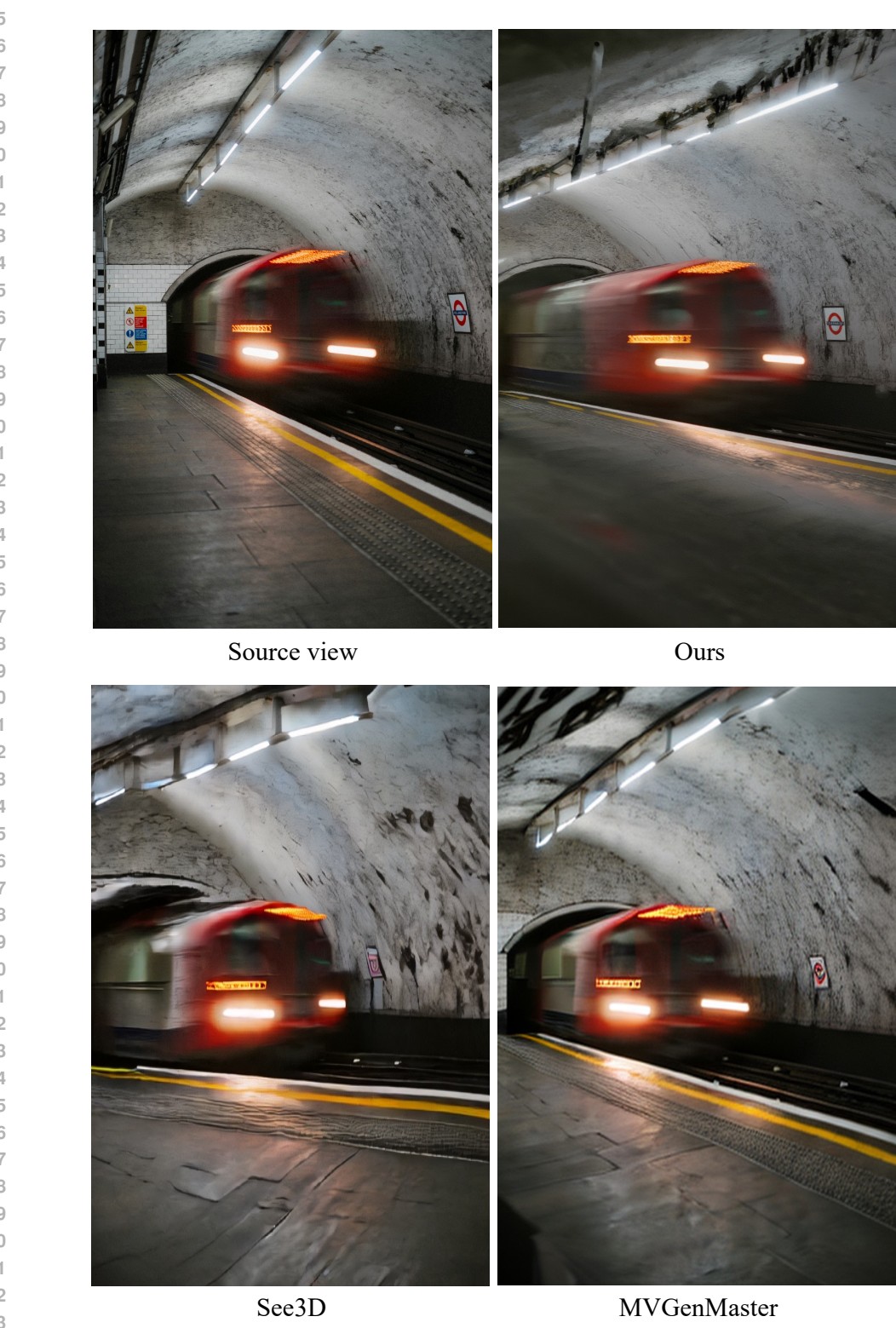

Figure 22: Supplementary novel view synthesis (NVS) examples on in-the-wild images. For the 30° leftward rotation, See3D distorts the scene, and MVGenMaster adds noise to the results.

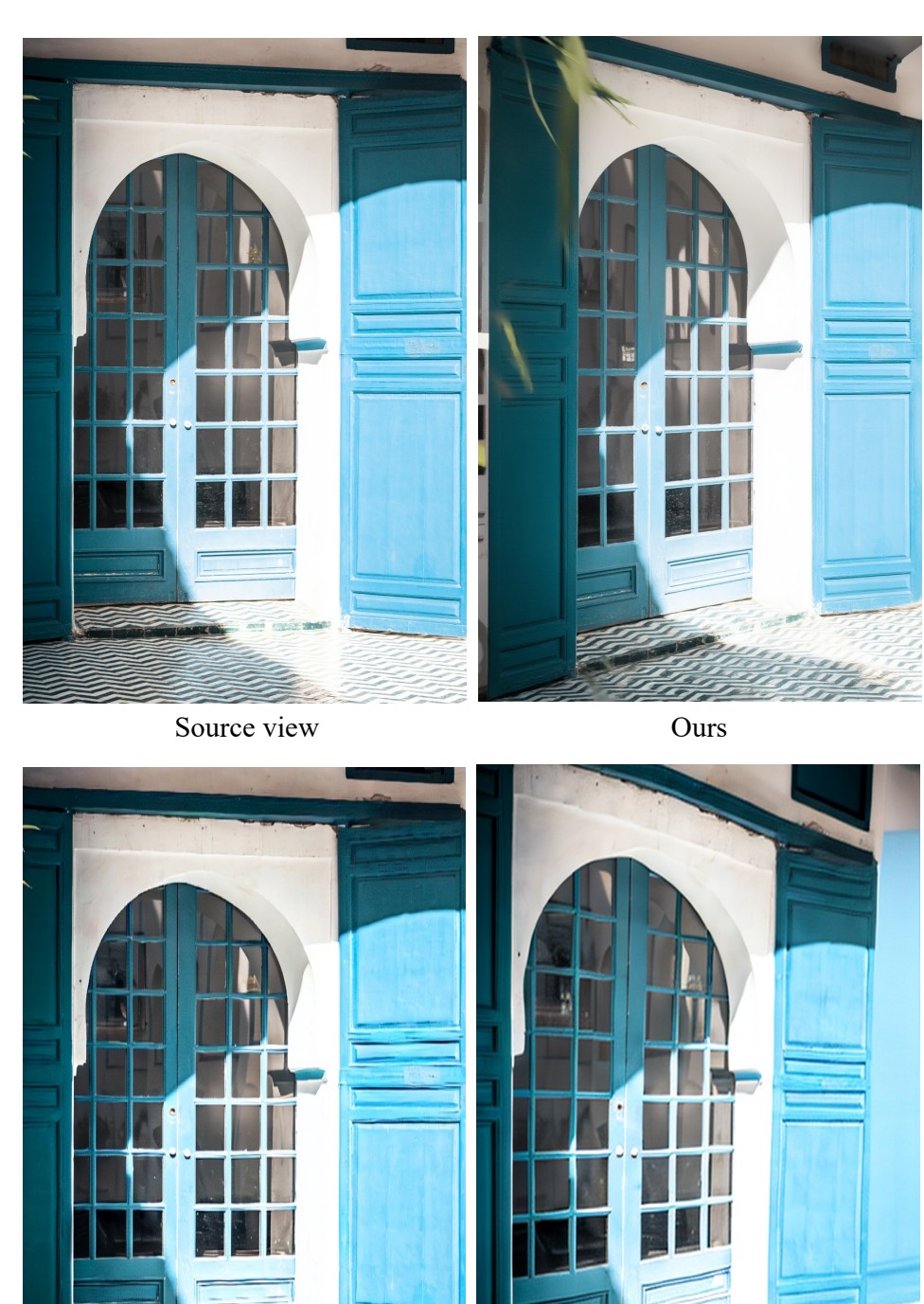

Source view        Ours

SEVA        FlexWorld

Figure 23: Supplementary novel view synthesis (NVS) examples on in-the-wild images. For the 30°
leftward rotation, SEVA shows little change, FlexWorld affects the floor patterns.

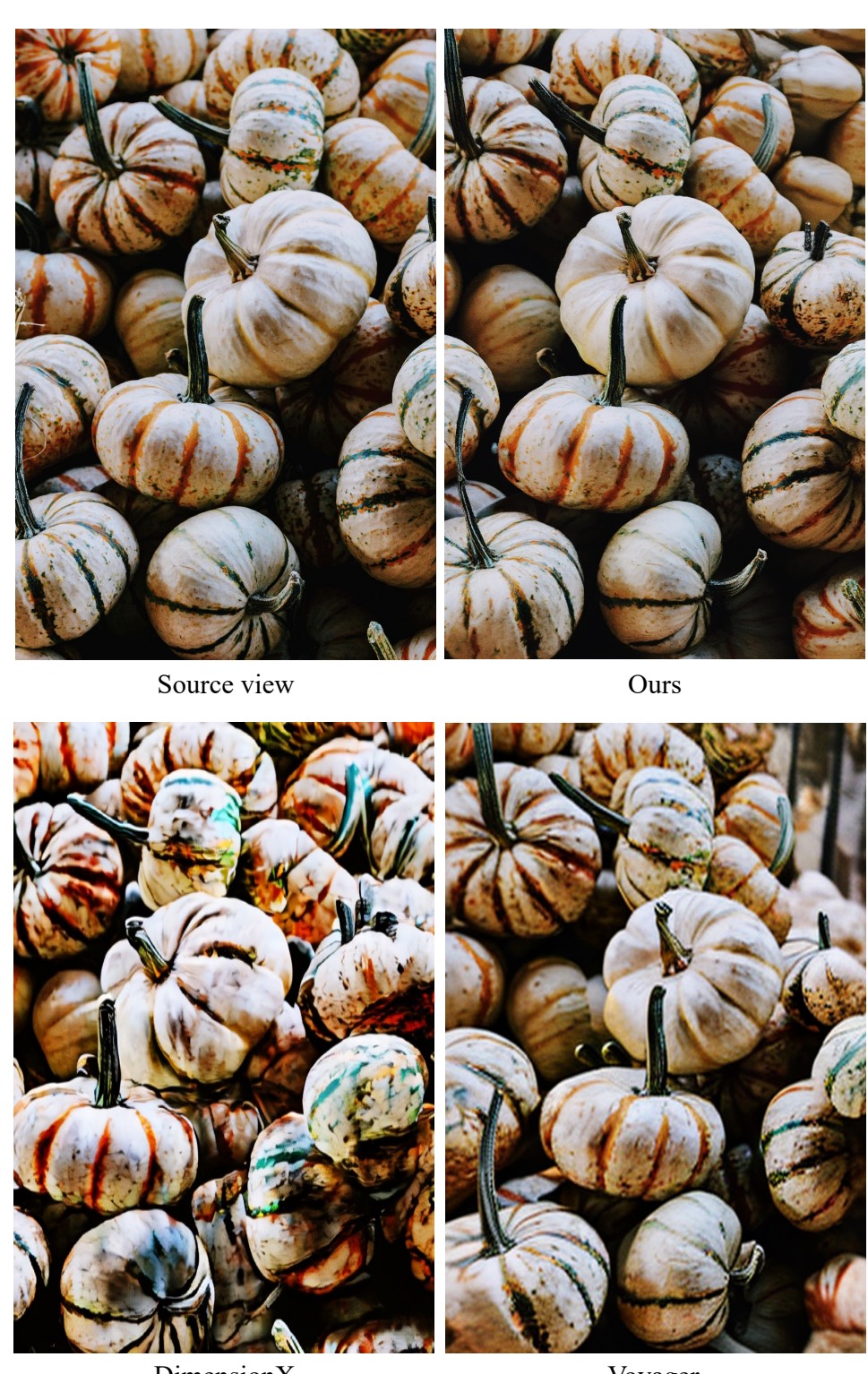

Figure 24: Supplementary novel view synthesis (NVS) examples on in-the-wild images. For the 30°
leftward rotation, DimensionX and Voyager alter both the object shapes and the overall image style.

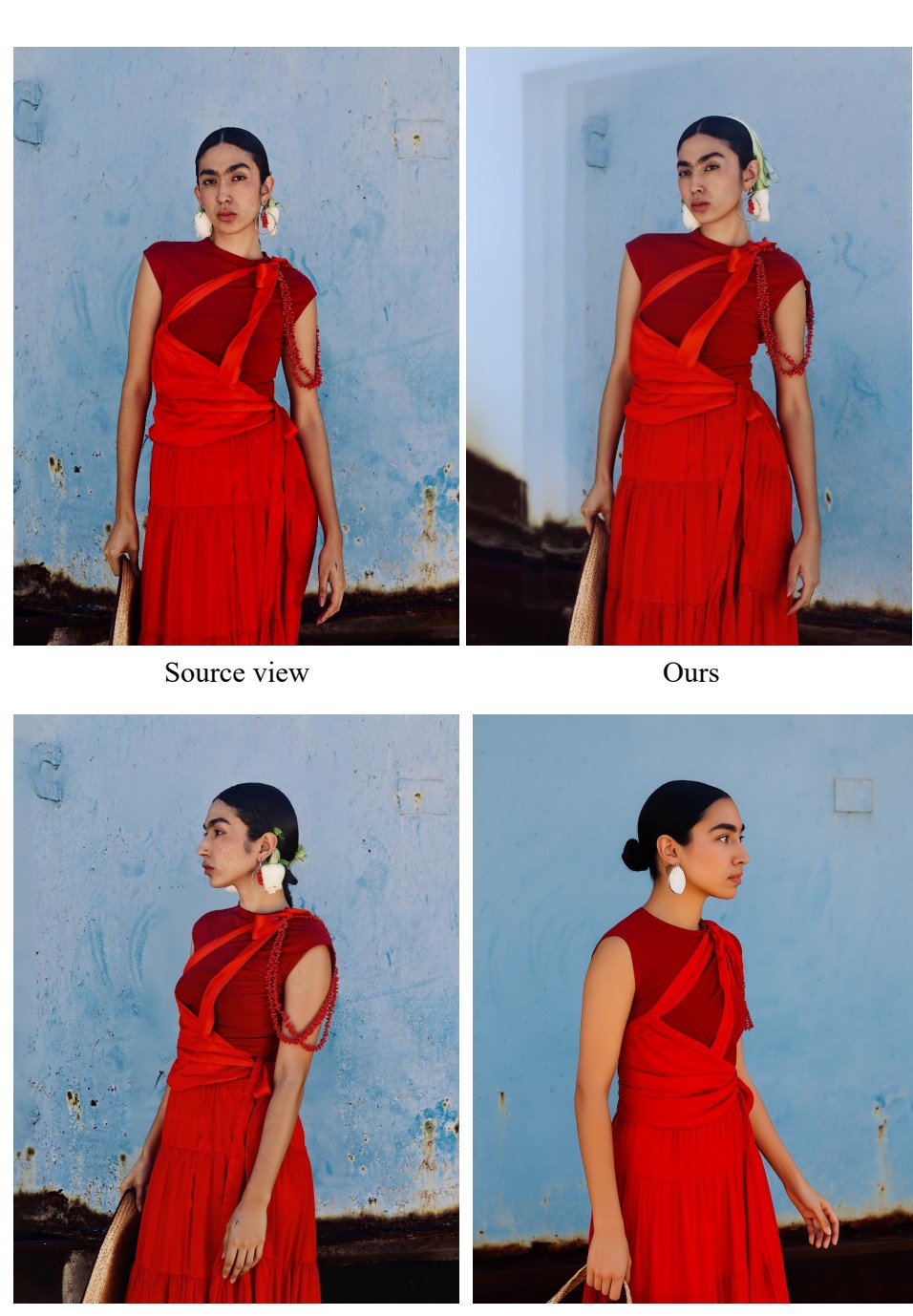

Source view · Ours

FLUX.1 Kontext · Qwen-Image-Edit

Figure 25: Supplementary novel view synthesis (NVS) examples on in-the-wild images. For the 30° rightward rotation, FLUX only rotates the person without rotating the background, Qwen rotates in the opposite direction and also changes the person's pose.

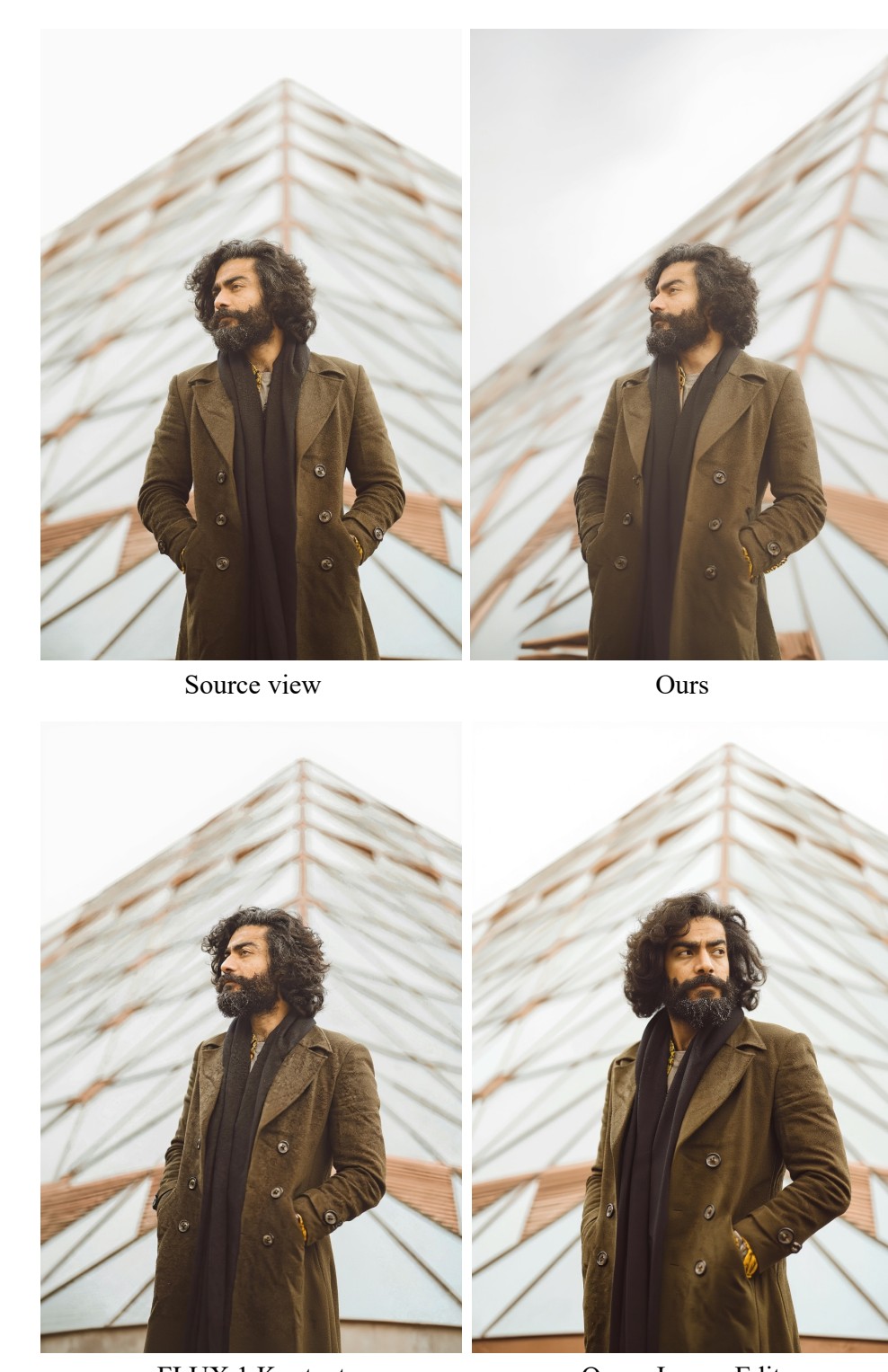

Figure 26: Supplementary novel view synthesis (NVS) examples on in-the-wild images. For the 30° rightward rotation, FLUX shows only minor changes without rotating the background, Qwen alters the person's pose and expression without rotating the background.

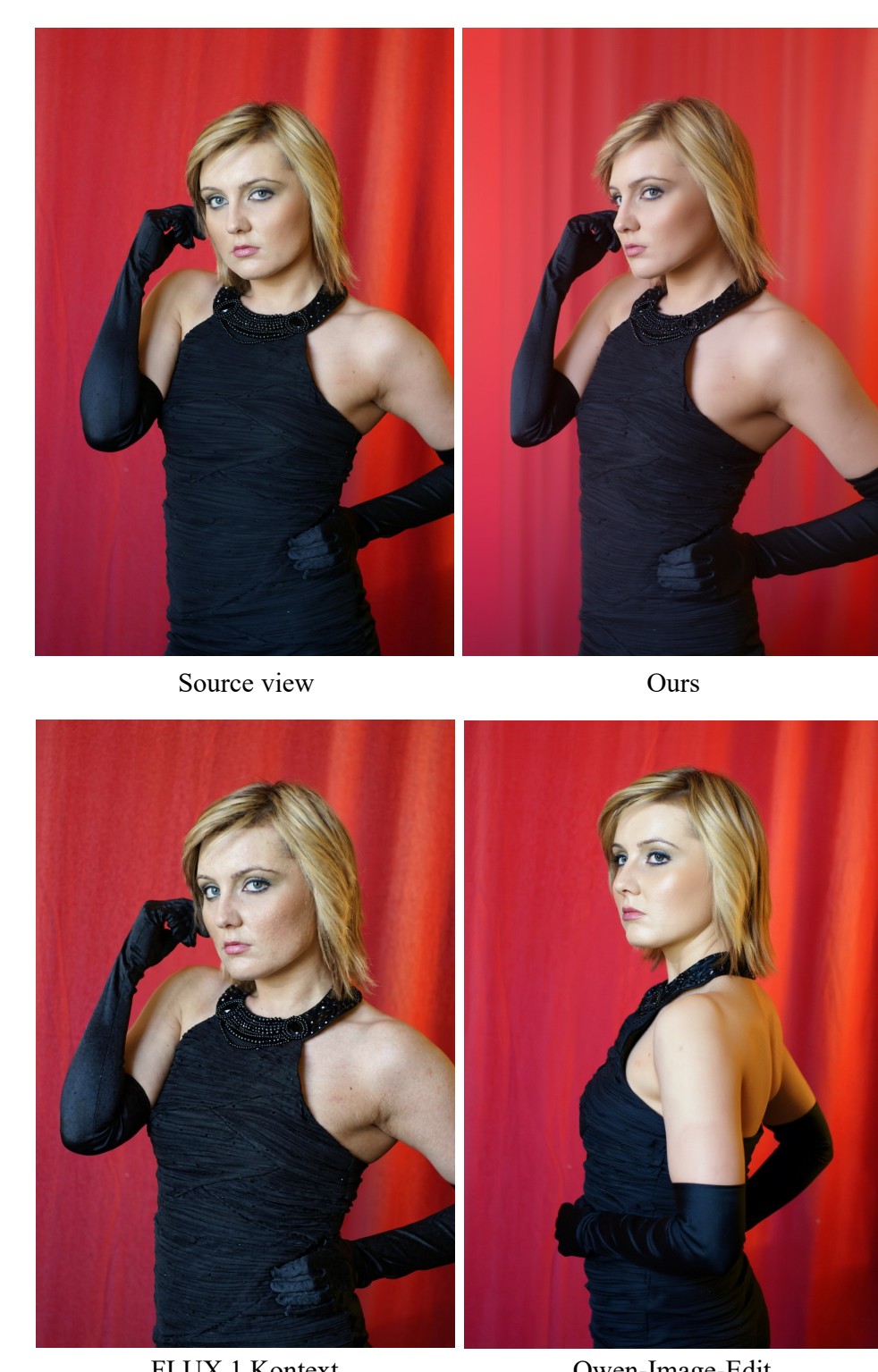

Source view — Ours

FLUX.1 Kontext — Qwen-Image-Edit

Figure 27: Supplementary novel view synthesis (NVS) examples on in-the-wild images. For the 30° rightward rotation, FLUX shows only minor changes, Qwen changes the person's pose without rotating the background.

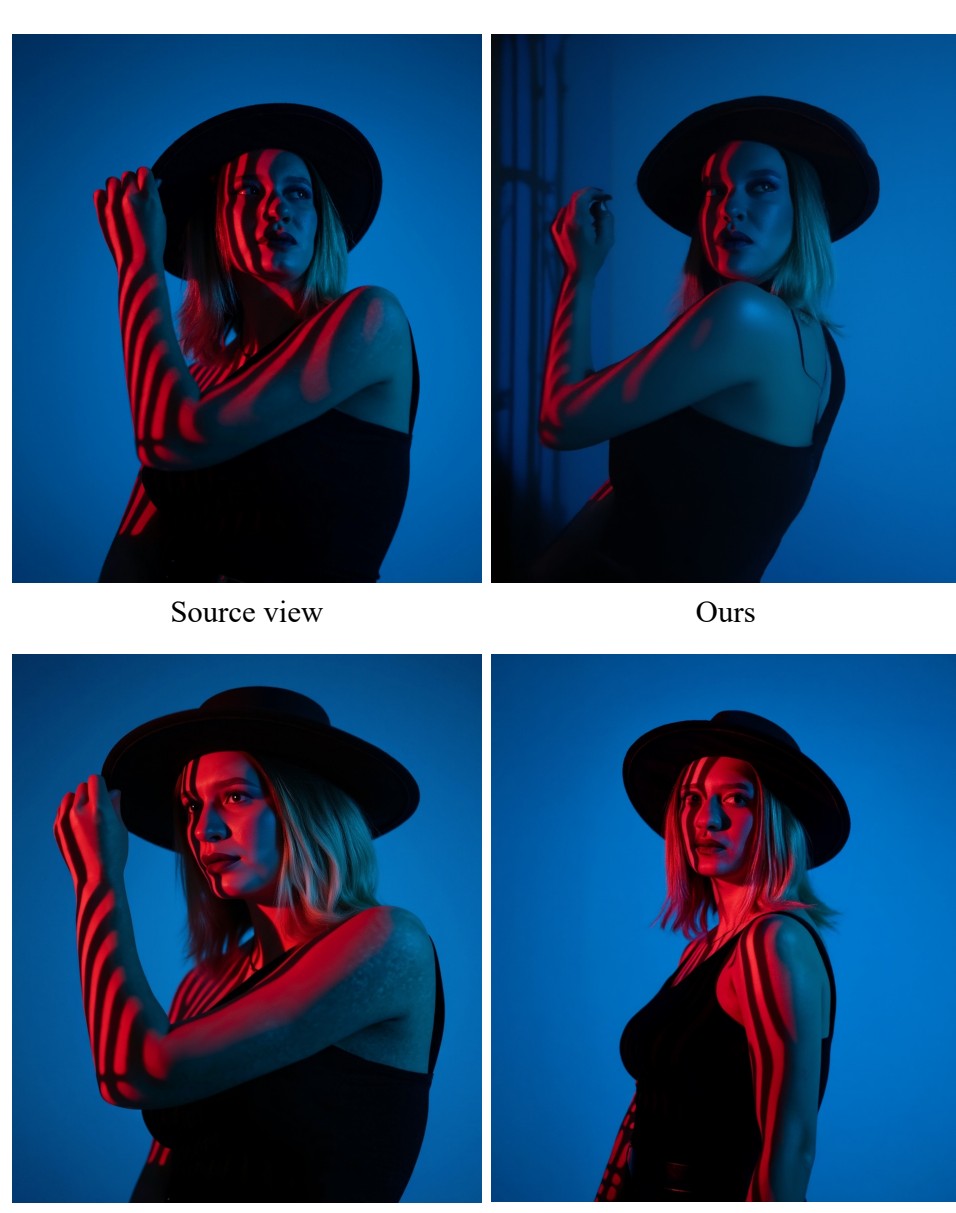

Figure 28: Supplementary novel view synthesis (NVS) examples on in-the-wild images. For the 30° rightward rotation, FLUX only rotates the person's head, Qwen changes the person's pose.

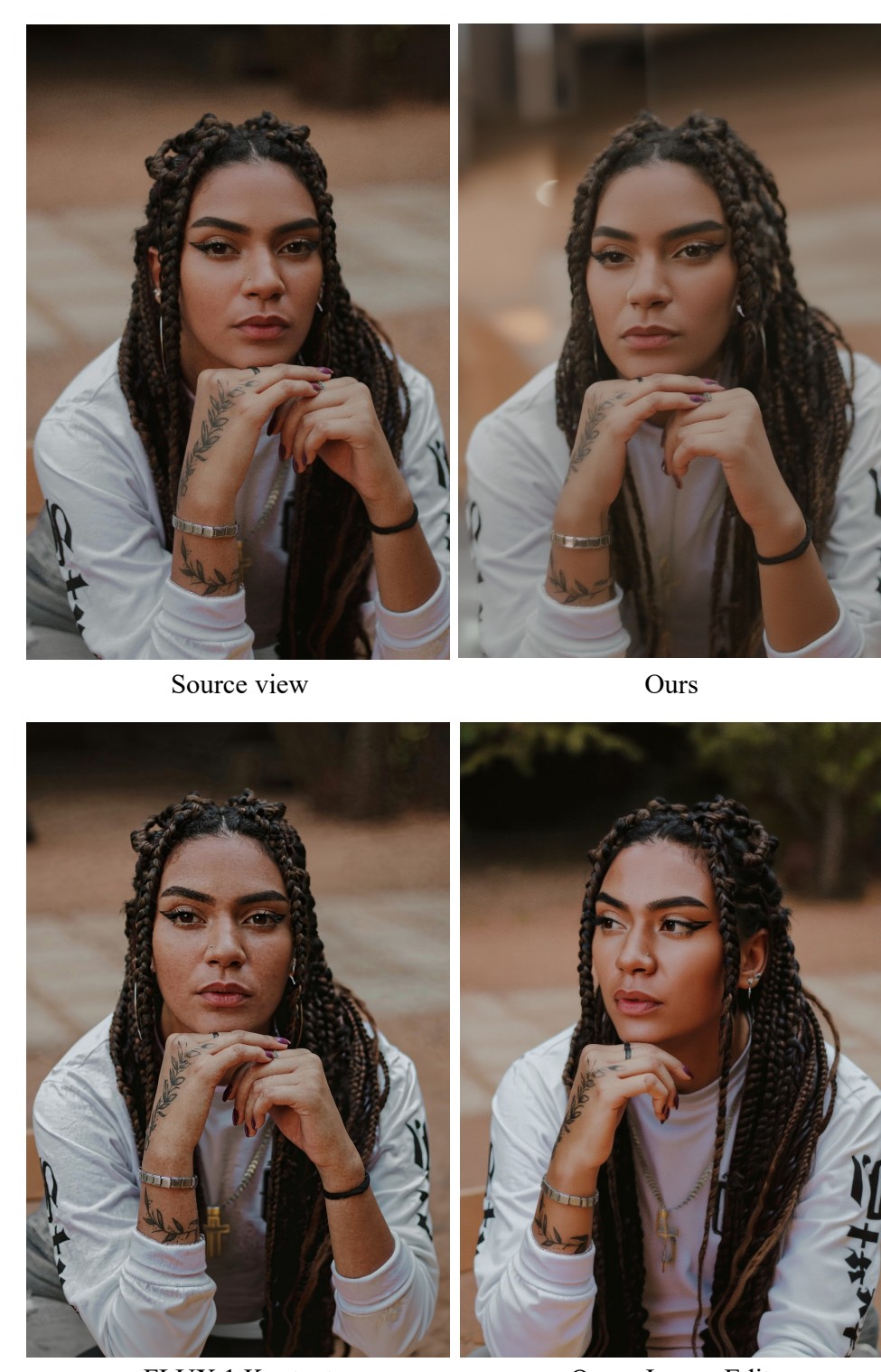

Figure 29: Supplementary novel view synthesis (NVS) examples on in-the-wild images. For the 30°
rightward rotation, FLUX shows only minor changes, Qwen changes the person's pose.

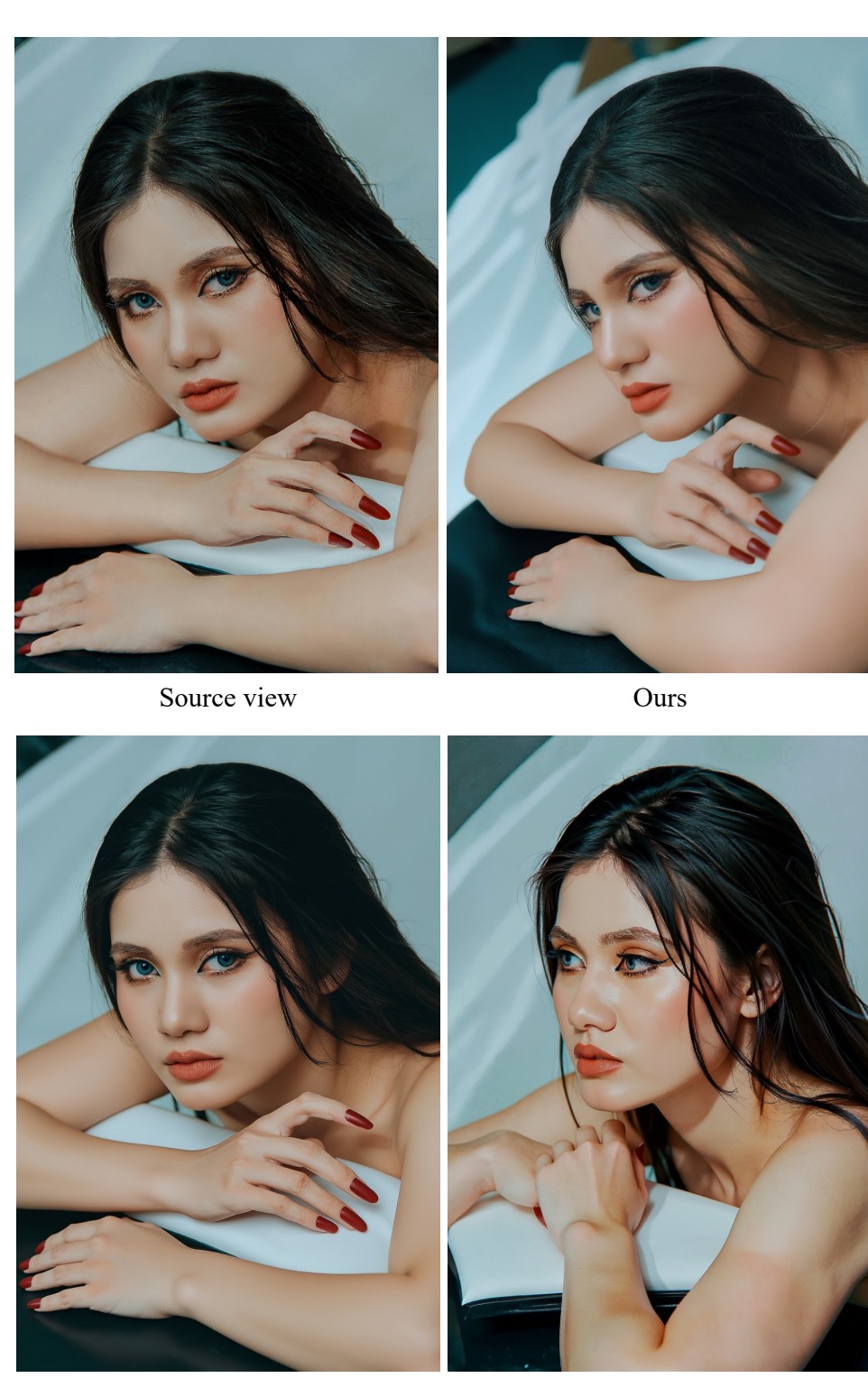

Source view          Ours

FLUX.1 Kontext          Qwen-Image-Edit

Figure 30: Supplementary novel view synthesis (NVS) examples on in-the-wild images. For the 30° rightward rotation, FLUX shows only minor changes, Qwen changes the person's pose.

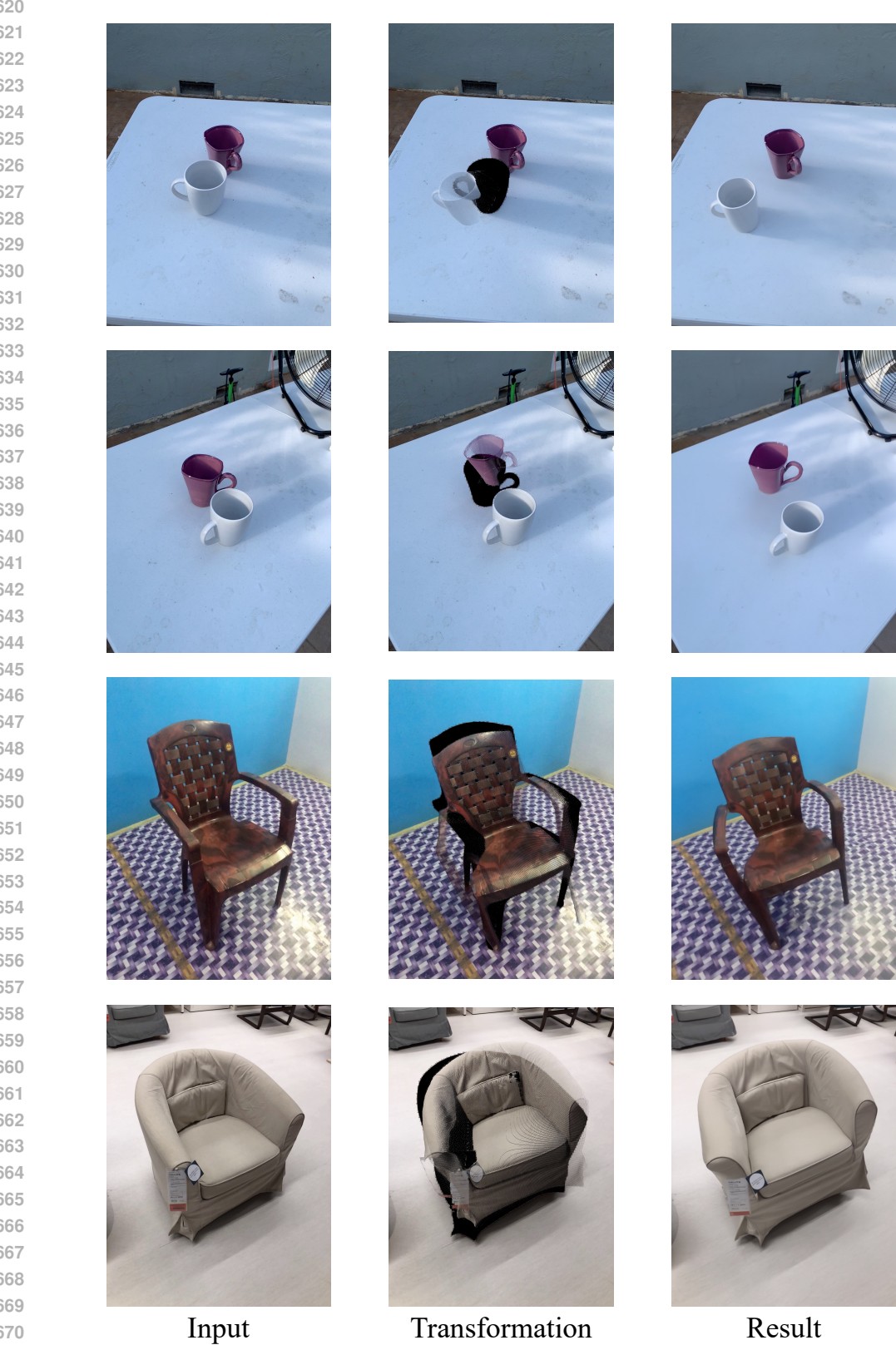

Input                    Transformation                    Result

Figure 31: Supplementary 3D-aware object editing examples on Objectron dataset.

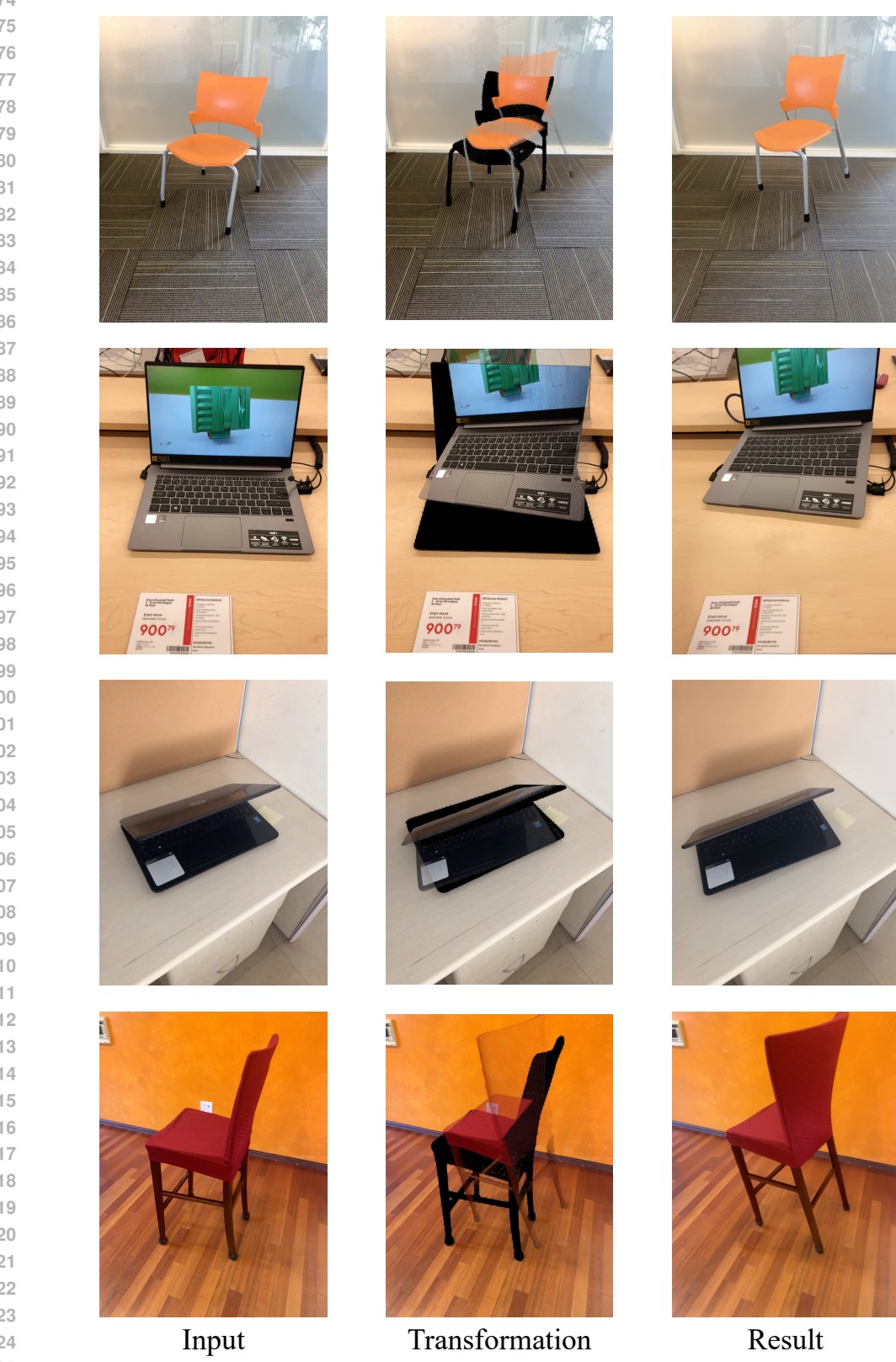

| Input | Transformation | Result |

Figure 32: Supplementary 3D-aware object editing examples on Objectron dataset.

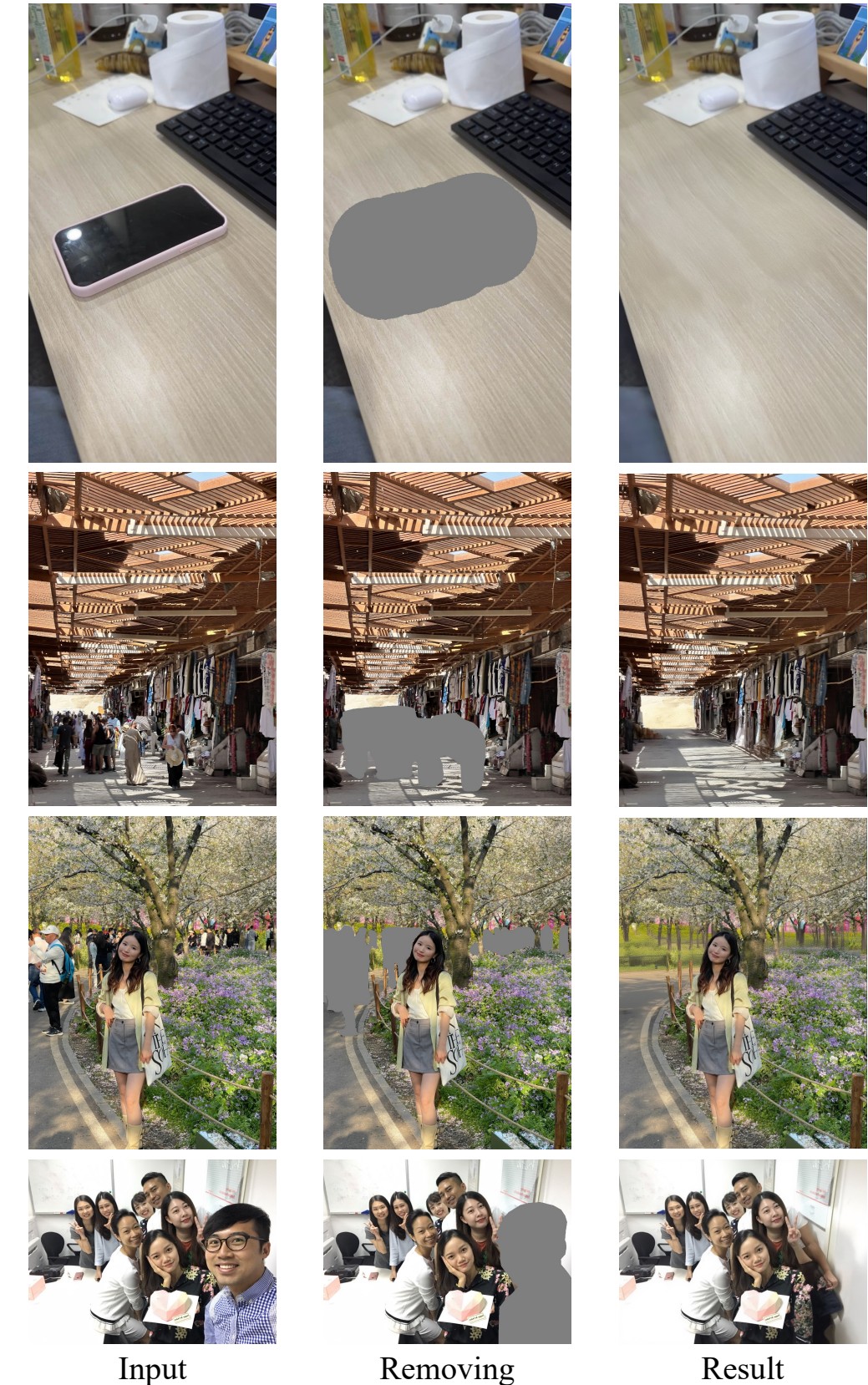

Figure 33: Supplementary object removing examples on in-the-wild images.

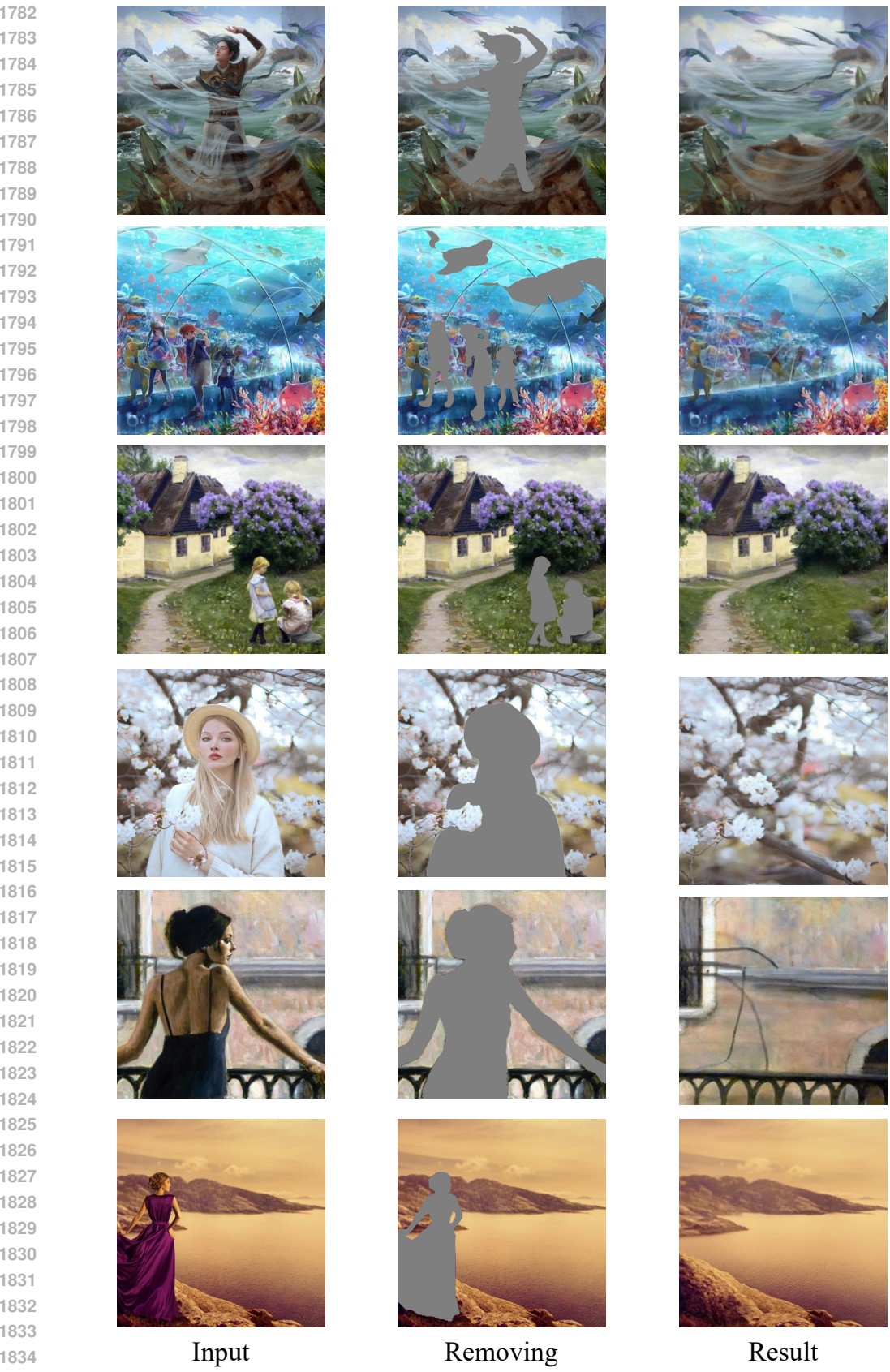

Input       Removing       Result

Figure 34: Supplementary object removing examples on in-the-wild images.