# OpenReview forum: "Positional Encoding Field"
_ICLR.cc/2026/Conference — ICLR 2026 Poster_

### Official Review · Reviewer_NKTZ · 2025-10-22

**Soundness:** 3
**Presentation:** 3
**Contribution:** 3
**Rating:** 6
**Confidence:** 4

**Summary:**

In this paper, the authors find that global spatial coherence comes from positional encodings (PEs) rather than token–token coupling, in which perturbing PEs preserves semantics while reorganizing layout. Based on this insight, the authors investigate the effect of PEs in the image generation process. And a new and neat positional encoding method is proposed that combines a 3D encoding (additional depth information) and a fine-grained patch encoding. With the proposed positional encoding manner, the PE-Field–augmented DiT achieves state-of-the-art performance on single-image novel view synthesis.

**Strengths:**

1. Interesting Concept Finding. The authors find that PEs control the spatial structure coherence during image generating or reconstructing process, motivating a new potential way for visual editing or synthesis.

2. Novel and Effective Positional Encoding. The authors consider the depth-aware information and more fine-grained semantics in the patch, then propose a new positional encoding that locates both the mentioned information.

3. Good Synthesis Result. With the proposed positional encoding, the introduced PE-augmented DiT achieves SOTA performance on single-image novel view synthesis.

**Weaknesses:**

1. Over-dependence on Monocular Reconstruction. As shown in the main framework, the monocular reconstruction provides essential 3D positional information for new view generation. How about the model’s robustness to terrible or noisy 3D position info injection?

2. Concerns about Multi-Level Positional Encoding. The head-layer match in fine-grained positional encoding is handcrafted in order.  More matching ways should be conducted to verify its robustness for fine-grained positional encoding.

3. More View Generation Scenes Discussion. 1) The visualization result from unseen view to seen view (i.e., view from left face to front face) should be reported. 2) Object-level or Region-level view synthesis.

4. Computing Cost Discussion. Since the proposed framework relies on monocular reconstruction, it may incur additional cost.

**Questions:**

1. Does the text-editing prompt influence the 3D-Positional encoding for view synthesis based on Flux Kontext?

2. Additionally, what is the instinct difference for text-control and PEs-control for view synthesis (visual editing and generation) during the training process?

---

> ### Author Response · Authors · 2025-11-21
>
> We appreciate your insightful questions and comments. Our responses are provided below.
>
> 1. Over-dependence on Monocular Reconstruction.
>
> Thank you for raising this insightful question.  In fact, compared to methods such as ViewCrafter that directly warp the input image, injecting the viewpoint transformation into the positional encoding (PE) is inherently more robust when the monocular reconstruction is inaccurate or noisy.
>
> Even if the depth prediction contains significant noise, our method **does not directly distort the input image (tokens)**. Instead, the model still receives a stable token grid and learns to produce an output that preserves coarse geometric structure while maintaining plausible visual content.
>
> We provide visualizations under extremely noisy reconstruction in **Appendix Fig. 14**, showing that the model continues to produce reasonable results despite severely corrupted depth inputs.
>
> 2. Concerns about Multi-Level Positional Encoding.
>
> Thank you for this excellent suggestion. We conducted additional experiments using several alternative fine-grained positional encoding assignment strategies between Transformer heads:
>
> i. **Reversed order:** we invert the ordering used in Figure 3.
>
> ii. **Grouped assignment:** Transformer heads are divided into four groups, each receiving one level-0 PE, one level-1 PE, and four level-2 PE channels. The groups are arranged in sequential order.
>
> iii. **Random assignment:** Randomly generate a positional-encoding assignment scheme.
>
>
> All three variants achieve **comparable performance** (±0.1 PSNR, ±0.01 SSIM, ±0.005 LPIPS ), demonstrating that our multi-level PE design is robust to different encoding assignment strategies.
>
> 3. More View Generation Scenes Discussion.
>
> We appreciate the reviewer’s request.  We have added the corresponding visualizations in **Appendix Fig. 15**.
>
>
> 4. Computing Cost Discussion.
>
> This is a fair concern.  Although monocular reconstruction introduces overhead, it is executed **only once per input image**, while diffusion sampling requires tens of steps. In practice, this overhead is marginal, contributing only **≈2–3%** of the total runtime. Additionally, depth estimation can be precomputed or cached offline, meaning it does not affect inference latency in real deployment scenarios.
>
> Furthermore, the methods we compare against—such as GenWarp, ViewCrafter, Gen3C, and MVGenMaster—**also require monocular reconstruction or equivalent geometry estimation**, so the computational setup remains consistent across baselines.
>
>
> 5. Does the text-editing prompt influence the 3D positional encoding for view synthesis based on Flux Kontext?
>
> This is a very interesting question.  When we fine-tune Flux Kontext as a novel-view generation model, the text tokens are effectively abandoned; we set the text input to either an empty string or a fixed placeholder. We experimented with reintroducing different prompts and found that **almost all prompts have negligible influence on the output**, confirming that text conditioning does not interfere with our 3D PE mechanism.
>
> 6. What is the intrinsic difference between text-control and PE-control for view synthesis (visual editing & generation) during training?
>
> PE-control operates in the **geometric space**, directly affecting the spatial arrangement of tokens based on camera pose, and the 3D structure inferred from monocular reconstruction.
>
> Because positional encodings participate directly in the attention computation, tokens from the source view and their corresponding locations in the target view obtain the most similar PE.  This provides an explicit and direct geometric guidance signal inside the Transformer.
>
> In contrast, for a text-control model, the Transformer must implicitly construct the correspondence between the source view and the target view through attention under different prompts, without any explicit geometric guidance.

---

### Official Review · Reviewer_o2gX · 2025-10-29

**Soundness:** 3
**Presentation:** 2
**Contribution:** 3
**Rating:** 2
**Confidence:** 3

**Summary:**

This paper presents a positional encoding field for novel view synthesis based on the DiT model. It is inspirted from the shuffling experiment and visualization of the position encoding for a standard model. The proposed method includes two parts: multi-level positional encodings for sub-patch detail modeling, and depth-aware extension to the rotary encoding. The proposed method has been evaluated on three datasets: Tanks-and-Temples, RE10K, and DL3DV. Both quantitative and qualitative experimental results demonstrate the effectiveness of the proposed method.

**Strengths:**

1.[effectiveness] The proposed method consistently outperforms the previous methods both qualitatively and quantitatively.

2.[motivation] The proposed method is well-motivated. It starts from an observation by shuffling the positional encodings. That helps the readers to better understand the motivation and design of the proposed method.

3.[ablations] Ablation studies has been carried out to demonstrate the effectiveness of individual components in the proposed method in section 4.3

4.[extension] The proposed method has probably more potential to be used in other spatial editing tasks as mentioned in section 4.4

**Weaknesses:**

1.[clarity] As described on L033-041, as well as figure 1, the proposed method is motivated by a position encoding shuffling experiment. What would be the worst case for a shuffled or re-ordered position encoding? How does that impact the model? If there is a threshold beyond which the generated output is completely messed up, the authors might have to rethink the connection between the position encoding shuffling expeirment and the proposed method.

2.[typesetting] The macro "\citep" or "\citet" should be used instead of a plain "\cite". The brackets are missing in manyplaces like the related works section, which makes it not easy to read.

3.[clarity] The caption of figure 1 is confusing. What does "perturbed" mean? The paper and the figure says "warp, shuffle, reorder". What is re-order again? Are they the same thing? Please use a consistent term to avoid confusion.

4.[clarity] On line 145, how are they re-ordered? Is it random?

5.[clarity] On line 246: how is the depth obtained during the training and testing process of the proposed method?

6.[design] The proposed method is only tested on the Flux model. Does it apply to other DiT-based models, such as the MMDiT (stable diffusion 3)? If so, the proposed method would be helpful for a wider community.

7.[fair comparison] In table 1, the proposed method outperforms the previous methods. But not every other method is built-on the Flux model. Therefore, it is unknown how much performance improvement originates from the Flux backbone itself. This would raise a fair comparison issue, and leave the real improvement of the proposed method in question.

**Questions:**

Generally this is a good paper. However the fair comparison issue makes it difficult to identify the real improvement made by the proposed method since it is mixed together with the powerful Flux backbone. Please see the questions in the weaknesses part.

---

> ### Author Response · Authors · 2025-11-21
>
> Thank you for the time and effort you invested in reviewing our paper. We have carefully considered your suggestions, and our detailed responses are as follows.
> 1. Clarification of the Position-Encoding Shuffling Experiment and Its Worst-Case Behavior.
>
> We clarify that the PE-shuffling experiment serves as an empirical observation.  The worst case corresponds to a fully random permutation (**Appendix Fig. 13**), which naturally destroys spatial continuity and breaks any DiT-like model—this is fully expected. However, our approach does not rely on PEs being arbitrarily re-orderable.
>
> What matters for our method is that the locally continuous PE reassignment we apply remains geometrically meaningful, and the DiT backbone responds to such perturbations in a stable, spatially aware manner. As a result, the spatial arrangement of the output tokens still follows a coherent reassignment pattern.
>
> This property is exactly what enables spatial-aware control directly through PEs:  the model can still infer relative spatial relationships because the reassigned PEs preserve coherent local neighborhoods rather than behaving like random noise.
>
> 2. Typesetting Issue.
>
> Thank you for pointing this out. We have revised all citations to correctly use \citep as suggested.
>
> 3. Clarification of Figure 1 Terminology.
>
> In the revised version, we unify all operations under the concept of **positional encoding reassignment**, where the PE originally assigned to position _i_ is reassigned to another position _j_ according to a permutation or warp-induced mapping.  This removes ambiguity between “warp,” “shuffle,” and “reorder.”
>
>
> 4. Clarification on How Position Encodings Are Re-Ordered.
>
> The Figure 1 reordering follows a structured **positional encoding reassignment** operation.  Concretely, a small cluster of tokens near the upper-left boundary is reassigned to slightly shifted locations toward the lower-right, while a symmetric group near the lower-right boundary is reassigned toward the upper-left.
>
> Just as illustrated by the two color-gradient maps in Figure 1, take the token located at the top-left corner of the image as an example.  Originally, it should be assigned to the 2D position corresponding to the top-left orange pixel in the first gradient map, i.e., the coordinate (0, 0).  After the reassignment, this token is shifted downward by _n_ units and rightward by _m_ units.
> Its new position becomes the top-left orange pixel in the second gradient map, corresponding to the coordinate _(m, n)_.
>
> Implementation-wise, we directly apply the the view transformation for another image and reuse that mapping on the current image (**see Appendix Fig. 13**).
>
>
> 5. Clarification on How Depth is Obtained During Training and Testing.
>
> As stated in Lines 312 and 321, we obtain depth for both training and testing using **VGGT**.
>
>
> 6. Applicability of the Proposed Method to Other DiT-Based Models (e.g., MMDiT / SD3)
>
> We have extended our method to the recently released **Qwen-Image-Edit** model, another MMDiT-based architecture.
> The results are comparable to those obtained with Flux-Kontext, demonstrating that our method generalizes well across DiT variants. This also partially addresses the reviewer’s fairness concern:  the backbone architecture has only a minor impact on the effectiveness of our PE formulation.
>
> 7. Fair Comparison Concern: Separating Improvements from Flux Backbone vs. Our Method
>
>
> This is a good question.  It is difficult to convert all prior methods into Flux-based versions for fully controlled comparisons, especially video-based NVS methods using strong temporal priors.
>
> To improve fairness, we added two **Flux-based baselines**:
>
> 1. **GenWarp-Flux:** a Flux implementation of the GenWarp pipeline.
>
> 2. **ViewCrafter-Flux:** a Flux variant of ViewCrafter/Gen3C, where the source image is warped using depth and Flux performs inpainting.
>
>
> Additionally, as suggested by Reviewer WUCM, we included a **training-free baseline** without any finetuning.  All these baselines show significantly worse performance than our approach, demonstrating that the gains come from **introducing view transformations into DiT positional encodings**. Corresponding visual comparisons are included in **Appendix Fig. 12**.
>
> **The comparison with these baselines is shown in the table below.**

---

> > ### Author Response · Authors · 2025-11-21
> >
> > | Method              |           | T & T     |           |           | RE10K     |           |           | DL3DV     |           |
> > | ------------------- | --------- | --------- | --------- | --------- | --------- | --------- | --------- | --------- | --------- |
> > |                     | PSNR↑     | SSIM↑     | LPIPS↓    | PSNR↑     | SSIM↑     | LPIPS↓    | PSNR↑     | SSIM↑     | LPIPS↓    |
> > | Training-Free Flux | 14.17     | 0.308     | 0.473     | 14.51     | 0.512     | 0.402     | 15.72     | 0.487     | 0.376     |
> > | GenWarp-Flux        | 17.51     | 0.587     | 0.313     | 17.42     | 0.658     | 0.324     | 17.02     | 0.584     | 0.335     |
> > | ViewCrafter-Flux   | 18.26     | 0.627     | 0.247     | 18.73     | 0.741     | 0.271     | 18.54     | 0.622     | 0.253     |
> > | Ours Qwen-image    | 22.31     | 0.741     | 0.167     | 21.27     | 0.808     | 0.171     | 22.61     | 0.748     | 0.153     |
> > | Ours Flux       | 22.12 | 0.732 | 0.174 | 21.65 | 0.816 | 0.162 | 22.23 | 0.742 | 0.154 |

---

### Official Review · Reviewer_WUCM · 2025-11-01

**Soundness:** 4
**Presentation:** 4
**Contribution:** 3
**Rating:** 6
**Confidence:** 4

**Summary:**

This paper builds upon a very interesting idea that DiT attention pattern is very very local. Starting from this idea, the authors gained the intution of editting the position/strcture/viewpoints of the image through editting the positional encoding of the DiT. The author conducted simple experiments in Figure-1 to show this concept.



Given these insights and polit study, the author developed more sophisticated designs for 3D-aware positional encodings. With this, the author builds upon flux-context (an open source image editting model) to implement his algorithm. The author finetuned the model on DL3DV MannequinChanllege with processed depth-map and camera poses.  After finetuning, the author showed impressive results with NVS on portaits images, object images and sceneray images.  Even though a simple call of the model can only generate relative view shifts, the author showed larger view-shift results by recurrsively applying the model.  When comparing with other methods, the author showed a higher PSNR.

**Strengths:**

1. The idea is quite neat. With polit study to build up intuitions, and interesting designs, and also good experiment design to validate the idea
2. The results are quite impressive! The visual quality is quite high. I do understand that each simple forward pass of the model can only edits a small view shifts. To address this, the author also showed recurssvely applied results, which is also quite cool.
3. The overall presentation is quite clear, and I can easily follow the logics of the author.

**Weaknesses:**

I lean towards accept the paper.

There are only several minor points, I don't think these are real weakness.

1. This methods is hard to be applied to multi-view novel view synthesis.  Maybe it's still possible is to build one single reference images by merging patches from multiple input views?
2. This demo would be even cooler if there are videos attached.  (Novel-view-synthesis videos with smoothly moving cameras.)

**Questions:**

Is it possible to build a training free baseline using the warpped 2D RoPE?  Cause from the intuition of Figure-1, it seems possible to build a training free baseline?  Correct me if I am wrong.

---

> ### Author Response · Authors · 2025-11-21
>
> We sincerely appreciate your recognition of the novelty and significance of our approach. Our responses to your questions are provided below.
>
> 1. This methods is hard to be applied to multi-view novel view synthesis. Maybe it's still possible is to build one single reference images by merging patches from multiple input views?
>
> Thank you for raising this point.  Our method can indeed be extended to multi-view novel view synthesis.  Models such as **VGGT** and **DUSt3R** can directly predict pointmaps and camera poses for multiple input frames.  Given these multi-view geometric estimates, we can:
>
> i. **Compute the target-view positions** for tokens originating from different views.
>
> ii. **Assign the corresponding 3D positional encodings (PEs)** to all tokens in the target frame.
>
> iii. **Feed all tokens jointly into the DiT backbone**, allowing the attention layers to fuse multi-view information.
>
>
> Conceptually, this is almost identical to the single-view setting.  We provide a two-view synthesis example in **Appendix Fig. 10**, demonstrating that our framework naturally generalizes to multiple input viewpoints.
>
>
>
> 2. This demo would be even cooler if there are videos attached.
>
> We completely agree that video demos with smoothly moving cameras would be visually compelling.  However, the current work is built on top of **image-based diffusion models**, and our main focus is on single-frame new-view generation.  Producing a long, smoothly interpolated video sequence requires an additional temporal-consistency mechanism, which is outside the scope of this work.
>
> Nonetheless, to approximate a moving-camera sequence, we include **a sequence of consecutively generated frames** in **Appendix Fig. 11**.  These results qualitatively show that the model can maintain geometric consistency across adjacent viewpoints, even without a temporal module.
>
>
> 3. Is it possible to build a training free baseline using the warpped 2D RoPE? Cause from the intuition of Figure-1, it seems possible to build a training free baseline?
>
> Thank you for this insightful suggestion. Indeed, Figure 1 motivates the possibility of constructing a **training-free baseline** by directly warping the 2D RoPE according to the predicted viewpoint transformation. Following your suggestion, we added such a baseline here and **Appendix Fig. 12** (qualitative results).
>
>
>
> | Method            |           | T & T     |           |           | RE10K     |           |           | DL3DV     |           |
> | ----------------- | --------- | --------- | --------- | --------- | --------- | --------- | --------- | --------- | --------- |
> |                   | PSNR↑     | SSIM↑     | LPIPS↓    | PSNR↑     | SSIM↑     | LPIPS↓    | PSNR↑     | SSIM↑     | LPIPS↓    |
> | Training-Free    | 14.17     | 0.308     | 0.473     | 14.51     | 0.512     | 0.402     | 15.72     | 0.487     | 0.376     |
> | **Ours Flux**     | **22.12** | **0.732** | **0.174** | **21.65** | **0.816** | **0.162** | **22.23** | **0.742** | **0.154** |
>
>
>
> This baseline provides a meaningful reference point, and we appreciate the reviewer for pointing out this direction.

---

> > ### Comment · Reviewer_WUCM · 2025-11-22
> > **post-rebuttal**
> >
> > Thanks for the reply, I read it and addressed most of my questions very clearly.

---

### Author Response · Authors · 2025-12-01
**Final Remarks**

We sincerely appreciate the reviewers for their time and for offering thoughtful and constructive feedback. Unfortunately, due to unexpected technical issues with OpenReview, we were unable to participate fully in the rebuttal phase, and we regret missing the opportunity for deeper engagement.

We are grateful that the reviewers recognized several strengths of our work, including its conceptual novelty, clear motivation, effective design, and strong performance. Their comments were highly encouraging and helped us further improve the paper:

1. **Strong and novel idea / motivation** Reviewers (WUCM, o2gX, NKTZ) agree the core idea is neat, well-motivated, and grounded in an insightful observation about positional encodings.

2. **Effective and novel positional encoding design** (o2gX, NKTZ) The new PE formulation is viewed as fresh and effective.

3. **Clear presentation and intuitive explanation** (WUCM, NKTZ) The logic and motivation are easy to follow and help readers understand the design choices.

4. **Strong experimental performance** (WUCM, o2gX, NKTZ) The method achieves high visual quality and SOTA results in single-image NVS, outperforming prior work.

5. **Potential for broader applications** (o2gX) The method likely extends to other spatial editing tasks beyond NVS.

In response to the reviewers’ suggestions, we have made several substantial revisions:

* **To further ensure fairness**, we added multiple new baselines built on the Flux architecture, including *GenWarp-Flux*, *ViewCrafter-Flux*, and a *Training-Free Flux* model, as well as experiments based on *Qwen-Image-Edit*. Taken together, the new baselines confirm that our method is not dependent on Flux-specific inductive biases and maintains its effectiveness across diverse pretrained architectures.

* We **expanded the discussion on monocular reconstruction**. While our method depends on monocular depth estimation, which can introduce noise and additional computation, this limitation is shared by many methods such as ViewCrafter and GenWarp. Importantly, our depth-aware positional encoding preserves the integrity of the original image tokens, giving our approach an advantage over methods that alter the latent representation more aggressively.

* We **added extensive visualizations** across diverse scenarios in Appendix Figures 10 through 15 to further illustrate the behavior and effectiveness of our method.

---

### Meta-Review · Area_Chair_1W8R · 2025-12-27

**Summary:**

The initial rating from the reviewers were 6, 2, and 6. During the discussion, Reviewer WUCM stated that the rebuttual addressed most questions. Other two reviewers did not have the chance to give the further comments. The AC read the paper, the reviews, and the rebuttal carefully. Please see the Reviewer Concerns session about the concerns and the AC's feedback.

**Reviewer Concerns:**

Reviewer WUCM

There are no big concerns from Reviewer WUCM. There are three suggestions: (1) extension to multi-view synthesis. (2) presenting video demos, and (3) building a training free baseline using the warped 2D RoPE.

The authors provided convincing feedbacks to the suggestions, by (1) providing  a two-view synthesis example; (2) giving a sequence of consecutively generated frames to approximate a moving-camera sequence; (3) adding the baseline results.

Reviewer o2gX

There are two major concerns (others are minor):
[design] The proposed method is only tested on the Flux model. Does it apply to other DiT-based models, such as the MMDiT (stable diffusion 3)? If so, the proposed method would be helpful for a wider community.

[fair comparison] In table 1, the proposed method outperforms the previous methods. But not every other method is built-on the Flux model. Therefore, it is unknown how much performance improvement originates from the Flux backbone itself. This would raise a fair comparison issue, and leave the real improvement of the proposed method in question.

The authors provided convincing feedback: presenting the recently released Qwen-Image-Edit model, another MMDiT-based architecture. The observations are consistent to  Flux-Kontext.
Regarding "fair comparison", the AC thinks the authors' feedback is convincing: "It is difficult to convert all prior methods into Flux-based versions for fully controlled comparisons, especially video-based NVS methods using strong temporal priors."


Reviewer NKTZ

The reviewer had a few concerns or suggestions about:
(1) Over-dependence on Monocular Reconstruction;
(2) Concerns about Multi-Level Positional Encoding;
(3) More View Generation Scenes Discussion;
(4) Computing Cost Discussion;
and two questions.
The authors provided detailed feedbacks. I was convinced by the feedbacks (Concern 1 and concern 3 are also raised by others)

**Reviewer Scores:**

Reviewer WUCM

The reviewer responded. The reviewer was satisfactory: most of the questions were addressed very clearly. The reviewer will be most likely to keep or upgrade the rating.

Reviewer o2gX

If the AC was the reviewer, the rating would be upgraded.

Reviewer NKTZ

The AC thinks that the reviewer will not downgrade the rating as the feedback is convincing.

---

### Decision · Program_Chairs · 2026-01-26

Accept (Poster)